# ADAPTIVE CURVATURE STEP SIZE: A PATH GEOMETRY BASED APPROACH TO OPTIMIZATION

## ABSTRACT

We propose the Adaptive Curvature Step Size (ACSS) method, which dynamically adjusts the step size based on the local geometry of the optimization path. Our approach computes the normalized radius of curvature using consecutive gradients along the iterate path and sets the step-size equal to this radius. The effectiveness of ACSS stems from its ability to adapt to the local landscape of the optimization problem. In regions of low curvature, where consecutive gradient steps are nearly identical, ACSS allows for larger steps. Conversely, in areas of high curvature, where gradient steps differ significantly in direction, ACSS reduces the step size. This adaptive behavior enables more efficient navigation of complex loss landscapes. A key advantage of ACSS is its adaptive behavior based on local curvature information, which implicitly captures aspects of the function's second-order geometry without requiring additional memory. We provide a generalized framework for incorporating ACSS into various optimization algorithms, including SGD, Adam, AdaGrad, and RMSProp. Through extensive empirical evaluation on 20 diverse datasets, we compare ACSS variants against 12 popular optimization methods. Our results consistently show that ACSS provides performance benefits. Our results consistently show that ACSS provides performance benefits. We provide PyTorch implementations of ACSS versions for popular optimizers at our anonymized code repository.

## 1 INTRODUCTION

Optimization algorithms are the canonical work-horses of machine learning, driving the process of finding optimal parameters for deep learning models (Soydaner, 2020; Kochenderfer & Wheeler, 2019; Beck, 2017). As model architectures grow in size and complexity, the efficiency of these algorithms becomes paramount. A key challenge is that the objective in many learning problems are inherently non-convex, often due to structural or data-related constraints that impose non-convexity (Jain et al., 2017). Such learning problems may induce intricate loss landscapes characterized by large tracts of low gradients interspersed with areas of steep gradients, presenting significant navigational challenges for optimization algorithms. Effective optimization methods must not only find good solutions but do so efficiently in terms of computation and memory usage, especially when dealing with large-scale models and datasets, where navigation on the loss landscape is likely to follow an intricate path (Anil et al., 2019).

In light of this, we propose a geometric path based solution to optimization: the Adaptive Curvature Step Size (ACSS) method. Our approach is motivated by the observation that the curvature of the optimization path itself contains information about the local geometry of the loss landscape. By utilizing this curvature information, we can incorporate second order information adaptively into the step size — without the need for explicit computation or storage of second-order derivatives, and without the need for careful tuning of learning rates.

The intuition behind ACSS is rooted in differential geometry. Specifically, the curvature of a path provides insight into how rapidly the gradient is changing, which is indicative of the local shape of the loss surface. In fact, the iterate path can be viewed as a finite-difference approximation to the gradient flow manifold. We note that the curvature of this manifold is a powerful proxy for the local geometry of the loss landscape. Our method, ACSS, implicitly captures information about the changing gradient, which is related to the Hessian. This provides some of the benefits of second-order methods while maintaining the computational efficiency of first-order approaches.

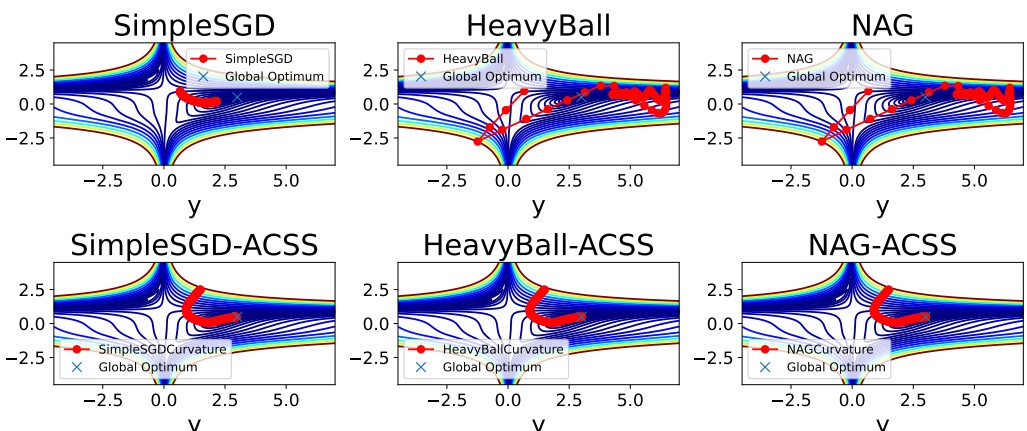

Figure 1: We plot the optimization paths of various optimizers on the Beale function which is characterized by steep valleys and a small area containing the global minimum. All optimizers start at $(-1.5, 2.5)$ with a learning rate of $1 \times 10^{-3}$. The function has a global minimum at $(3, 0.5)$; The ACSS versions of the optimizers converge here, without the use of any additional memory to store higher order moments.

## 1.1 RELATED WORKS:

**First Order Methods:** While first-order methods like Stochastic Gradient Descent (SGD) have low memory requirements, they converge slowly, particularly in ill-conditioned problems (Tian et al., 2023). Momentum based methods such as HeavyBall and NAG dampen oscillations to a certain degree (Sra et al., 2012; Nesterov, 2013), yet have limited ability to adapt when the loss landscape requires a change in direction of iterate (as seen in Figure 1).

**Variance of Gradient:** To address the limitations of basic SGD, several adaptive methods that adjust learning rates based on gradient statistics have been proposed. Adagrad accumulates squared gradients to adaptively tune learning rates, but it suffers from an ever-decreasing learning rate (Duchi et al., 2011). RMSProp improves upon this by using an exponentially decaying average of squared gradients, maintaining a more stable learning rate over time (Hinton et al., 2012). Adam and its variants (Kingma & Ba, 2014) further incorporate momentum, combining the benefits of adaptive learning rates and momentum to achieve better performance in various scenarios. AdamW enables better generalization through through weight decay regularization Loshchilov & Hutter (2017). AMSGrad addresses the convergence issues of Adam by ensuring that the learning rate does not increase, thereby providing better theoretical guarantees and more stable convergence in practice (Reddi et al., 2019). Nadam, and its weight decay variant NAdamW, integrate Nesterov momentum into the Adam framework, leading to faster convergence by anticipating the future position of

| Optimizer | Weights | Gradients | Momentum | Accumulated Squared Gradients | Exp. Avg. of Gradients | Exp. Avg. of Squared Gradients |
|---|---|---|---|---|---|---|
| SimpleSGD | ✓ | ✓ | ✗ | ✗ | ✗ | ✗ |
| HeavyBall | ✓ | ✓ | ✓ | ✗ | ✗ | ✗ |
| NAG | ✓ | ✓ | ✓ | ✗ | ✗ | ✗ |
| Adagrad | ✓ | ✓ | ✗ | ✓ | ✗ | ✗ |
| RMSProp | ✓ | ✓ | ✗ | ✓ | ✗ | ✗ |
| Adadelta | ✓ | ✓ | ✗ | ✗ | ✗ | ✓ |
| Adam | ✓ | ✓ | ✗ | ✗ | ✓ | ✓ |
| AdamW | ✓ | ✓ | ✗ | ✗ | ✓ | ✓ |
| AMSGrad | ✓ | ✓ | ✗ | ✗ | ✓ | ✓ |
| NAdam | ✓ | ✓ | ✗ | ✗ | ✓ | ✓ |
| NAdamW | ✓ | ✓ | ✗ | ✗ | ✓ | ✓ |
| RMSPropMomentum | ✓ | ✓ | ✓ | ✓ | ✗ | ✗ |

Table 1: Memory requirements for different optimizers during backpropagation

the parameters (Dozat, 2016). However, these adaptive methods are not without drawbacks. They can sometimes lead to poor generalization (Wilson et al., 2017), and the implicit learning rate decay inherent in their designs can cause convergence issues in some scenarios (Reddi et al., 2019). Moreover, lack the ability to fully capture and utilize the local geometric information of the loss landscape, and often require careful tuning of hyper-parameters. We provide a study on the memory requirements of various optimizers in terms of the number of parameters in the model, in Table 1.

**Second Order Methods:** Second-order optimization methods typically offer better convergence properties, but Hessian based methods can get prohibitively expensive (Anil et al., 2020). Works like Gupta et al. (2018); Goldfarb et al. (2020); Singh et al. (2023) exploit the structure of the neural architecture that is being optimized (using factoring over layers) to reduce the computational cost, but these can face numerical instabilities. Subsequent works like Sophia (Liu et al., 2023) and AGD (Yue et al., 2023) address these issues, and yet have memory overhead. Recent works like Feinberg et al. (2024); Yen et al. (2024) address the memory issue to a certain degree, but they are essentially approximating the preconditioning tensor, which has a computation cost. Still other methods like VeLO (Metz et al., 2022) are frameworks that decide the optimization parameters using a small neural network — which has a wall-clock time overhead.

## 1.2 OUR CONTRIBUTIONS

1. **Novel Optimization Approach:** We introduce the Adaptive Curvature Step Size (ACSS) method, a new optimization algorithm that leverages the geometric properties of the optimization path to dynamically adjust step sizes. ACSS incorporates local curvature information derived from consecutive gradients, providing benefits typically associated with higher-order methods while maintaining the computational efficiency of first-order approaches. This approach allows ACSS to adapt to the local landscape of the optimization problem automatically, eliminating the need for careful manual tuning of step sizes typically required in traditional optimization methods.

2. **Low Memory Footprint with Performance Benefits:** Unlike many optimization methods that require significant additional memory for storing pre-conditioners or momentum terms, ACSS offers second-order benefits while maintaining the memory footprint of the base optimizer. Our experiments demonstrate that ACSS variants, particularly for optimizers like SGD, HeavyBall, and NAG that do not store squared gradients, show significant performance improvements across diverse datasets. For instance, SimpleSGD-ACSS often outperforms more complex methods like AdamW and AMSGrad, despite its lower memory requirements. This makes ACSS particularly suitable for large-scale optimization problems, where the reduced memory footprint can be leveraged to increase the number of parameters being optimized.

3. **Theoretical Foundation:** We provide a comprehensive theoretical analysis of ACSS, proving bounds on effective step size, stability under perturbations, convergence rates for strongly convex functions, and scale invariance properties. This analysis demonstrates ACSS's adaptive behavior to local curvature and offers insights into its relationship with both first-order and second-order optimization techniques.

4. **PyTorch Implementation:** To facilitate adoption and further research, we provide efficient PyTorch implementations of the ACSS variants for popular optimizers, at our anonymized GitHub repository, making it easy to incorporate our method into existing machine learning workflows and reproduce our results.

In the next section, we provide the necessary notations and theoretical machinery for ACSS.

## 2 NOTATIONS AND METHOD

Consider a function $f : \mathbb{R}^n \times \mathcal{D} \to \mathbb{R}$ that we wish to minimize with respect to its first argument $w \in \mathbb{R}^n$. The optimization path traced by iterates $\{w_t\}$ can be viewed as a discrete approximation of a continuous curve in parameter space. Let $w_t \in \mathbb{R}^n$ be the parameter at iteration $t$, and $g_t = \nabla_w f(w_t, \mathcal{B}_t)$ be the gradient computed using a batch $\mathcal{B}_t \subset \mathcal{D}$.

In differential geometry, the curvature $\kappa(s)$ of a curve $w(s)$ parameterized by arc length $s$ is defined as:

$$\kappa(s) = \left\| \frac{dT(s)}{ds} \right\|, \tag{1}$$

where $T(s) = \frac{dw(s)}{ds}$ is the unit tangent vector. The radius of curvature is given by $\rho(s) = \frac{1}{\kappa(s)}$.

To relate this to our discrete optimization steps, we approximate the curvature using finite differences. Let $\eta$ be the base learning rate, and $g_t' = \nabla_w f(w_t - \eta g_t, \mathcal{B}_t)$ be the gradient at a *tentative* next point. We define the normalized radius of curvature as:

$$r_t := \frac{\|g_t\|}{\|g_t - g_t'\|}. \tag{2}$$

This approximation allows us to estimate the local curvature of the loss landscape without explicitly computing second-order derivatives.

To ensure numerical stability, we introduce a cap on the normalized radius of curvature:

$$\hat{r}_t := \min\{r_{\max}, r_t\}, \tag{3}$$

where $r_{\max}$ is the maximum allowed curvature.

**Update Rule:** Incorporating this adaptive curvature step size, we define the update rule as:

$$w_{t+1} := w_t - \eta \times \hat{r}_t \times \frac{g_t}{\|g_t\|} \quad (\textbf{Eq. 1}) \tag{4}$$

This update can be interpreted as moving in the direction of the negative gradient $\frac{g_t}{\|g_t\|}$ with a step size dynamically adjusted by $\eta \times \hat{r}_t$ based on the local curvature of the loss landscape.

The proposed Adaptive Curvature Step Size (ACSS) method aims to balance the trade-off between convergence speed and stability by adapting the step size according to the geometry of the optimization path. In regions of low curvature, it allows for larger steps to accelerate progress, while in highly curved areas, it reduces the step size to maintain stability.

## 2.1 Algorithm

We now provide this update rule in the form of an Algorithm.

---

**Algorithm 1:** Stochastic gradient descent with adaptive curvature step size (SGD-ACSS)

---

**Input:** Function $f_w : \mathcal{D} \to \mathbb{R}$, initial parameters $w_0 \in \mathbb{R}^n$, base learning rate $\eta$, maximum
      radius $r_{max}$, number of iterations $T$, batch size $B$
**Output:** Optimized parameters $w_T$
**for** $t = 0$ **to** $T - 1$ **do**
    Sample a mini-batch $\mathcal{B}_t$ from $\mathcal{D}$;
    Compute gradient $g_t = \nabla f_w(w_t; \mathcal{B}_t)$;
    Compute tentative next point gradient $g_t' = \nabla f_w(w_t - \eta g_t; \mathcal{B}_t)$;
    Compute normalized radius of curvature $r_t = \frac{||g_t||}{||g_t - g_t'||}$;
    Compute capped radius $\hat{r}_t = \min\{r_{max}, r_t\}$;
    Update parameters $w_{t+1} = w_t - \eta \times \hat{r}_t \times \frac{g_t}{||g_t||}$;
**end**
**return** $w_T$

---

## 3 Theoretical Analysis

We provide theoretical guarantees for the Adaptive Curvature Step Size (ACSS) method. Our analysis focuses on the method's convergence properties, step size bounds, and adaptive behavior. Detailed proofs for all theorems can be found in the Appendix Section B.

### 3.1 STEP SIZE BOUNDS AND CONVERGENCE

We begin by establishing bounds on the effective step size of ACSS and proving its convergence for strongly convex functions.

**Theorem 1** (Bounded Step Size of ACSS). Let $f : \mathbb{R}^n \to \mathbb{R}$ be an $L$-smooth and $\mu$-strongly convex function. Consider the ACSS update rule with $r_{\max} \leq \frac{2}{\eta(\mu+L)}$. Then, the effective step size $\eta_{\text{eff}} = \eta \hat{r}_t$ is bounded as follows:

$$\frac{1}{L} \leq \eta_{\text{eff}} \leq \frac{2}{\mu + L}$$

for all iterations $t$.

This theorem ensures that ACSS maintains step sizes within a range that promotes stable convergence. Building on this result, we establish the convergence rate for ACSS:

**Theorem 2** (Convergence Rate for ACSS on Strongly Convex Functions). Let $f : \mathbb{R}^n \to \mathbb{R}$ be an $L$-smooth and $\mu$-strongly convex function. Under the ACSS update rule, for all $t \geq 0$:

$$\|w_t - w^*\|^2 \leq \left(1 - \frac{\mu^2}{L^2}\right)^t \|w_0 - w^*\|^2.$$

This theorem indicates that ACSS achieves linear convergence for strongly convex functions, with a rate comparable to standard gradient descent methods.

It is important to note that while the theoretical results presented in this section are derived for the deterministic gradient setting, the empirical results of ACSS, as discussed in Section 4, involves its use in stochastic settings with mini-batch optimization. The extension of these theoretical guarantees to the stochastic case is a potential area for future work. Nevertheless, our analysis does extend to scenarios involving bounded gradient perturbations, as detailed in the following subsection.

### 3.2 STABILITY UNDER PERTURBATION

Next, we present results on the stability of ACSS under gradient perturbations and its convergence guarantees for L-smooth and $\mu$-strongly convex functions.

**Theorem 3** (Stability of ACSS Under Gradient Perturbations). Let $f : \mathbb{R}^n \to \mathbb{R}$ be an $L$-smooth and $\mu$-strongly convex function. Assume the gradients are perturbed such that $\tilde{g}_t = g_t + \delta_t$ and $\tilde{g}'_t = g'_t + \delta'_t$, where $\|\delta_t\| \leq \varepsilon$ and $\|\delta'_t\| \leq \varepsilon$ for some $\varepsilon > 0$. Then, the difference between the updates using exact and perturbed gradients satisfies:

$$\|\tilde{w}_{t+1} - w_{t+1}\| \leq \frac{4\eta_{\max}\varepsilon}{m - \varepsilon},$$

where $\eta_{\max} = \frac{2}{L+\mu}$ and $m$ is a lower bound on the gradient norm.

While this theoretical result provides partial insights under specific assumptions, it may not fully capture ACSS's behavior in complex, non-convex landscapes. However, our extensive experiments in Section 4 may provide further evidence of ACSS stability properties across several difficult-to-optimize problems and diverse common machine learning datasets.

### 3.3 ADAPTIVE BEHAVIOR AND SCALE INVARIANCE

Finally, we examine the scale invariance property of ACSS.

**Theorem 4** (Scale Invariance of ACSS Effective Step Size). For any scalar $\alpha > 0$, scaling the base step size $\eta$ by $\alpha$ results in the same parameter updates for quadratic functions and approximately the same updates for general $L$-smooth and $\mu$-strongly convex functions, assuming $r'_t \leq r_{\max}$.

This scale invariance property suggests that ACSS is not sensitive to the choice of base step size — a significant practical advantage. ACSS automatically adapts its effective step size to the local geometry of the loss landscape, taking larger steps in low-curvature regions and smaller steps in high-curvature areas. This behavior mitigates the need for manual step size tuning and allows ACSS to maintain near-optimal convergence rates across varying landscapes without requiring prior knowledge of function-specific parameters. In contrast, SGD often requires careful manual tuning of step sizes to achieve similar convergence rate guarantees, which is challenging, particularly when optimizing functions with varying curvature across the parameter space.

# 4 EXPERIMENTS

## 4.1 CROSS-DATASET PERFORMANCE ANALYSIS OF ACSS

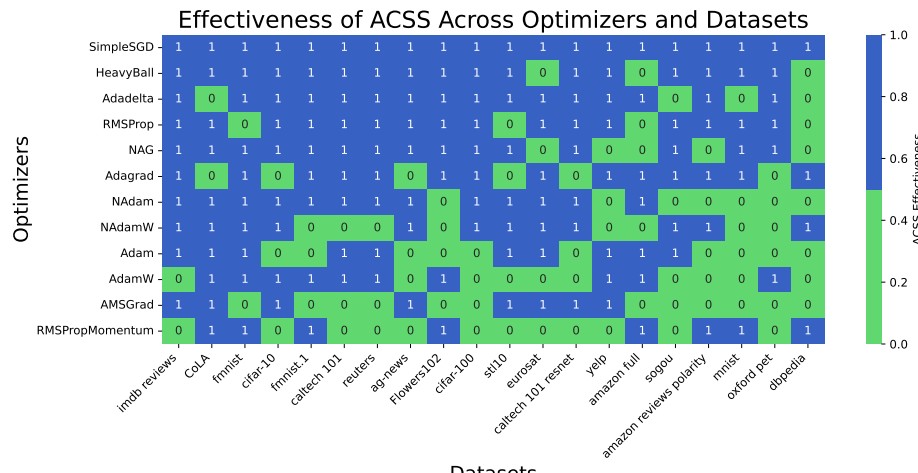

Figure 2: Binary representation of ACSS effectiveness across datasets and optimizers. Values indicate improvement (1) or no improvement (0) in training loss after a fixed number of epochs.

Figures 2 and 3 present a comprehensive evaluation of ACSS across 12 optimizers and 20 diverse datasets. ACSS demonstrates consistent performance improvements for most optimizer-dataset combinations. Significantly, SimpleSGD exhibits the most robust improvement across all datasets.

Optimizers that do not inherently use second-order information show the highest improvements, suggesting that ACSS effectively incorporates second-order information through loss landscape topology. SGD, HeavyBall, and NAG demonstrated mean training loss improvements of approximately 0.5 across 20 datasets using their respective ACSS versions.

Vision-related benchmarks, including Caltech 101, CIFAR-100, Flowers102, and STL10, showed the most significant improvements. The 18-layer ResNet variant exhibited the best performance, while the MNIST dataset with a simple neural network showed less pronounced improvements, likely due to the inherent effectiveness of most optimizers on simpler models.

**Key Takeaways:** ACSS provides improvements for most optimizers across various datasets. In cases where regular versions outperform ACSS, the difference in training loss is typically minimal.

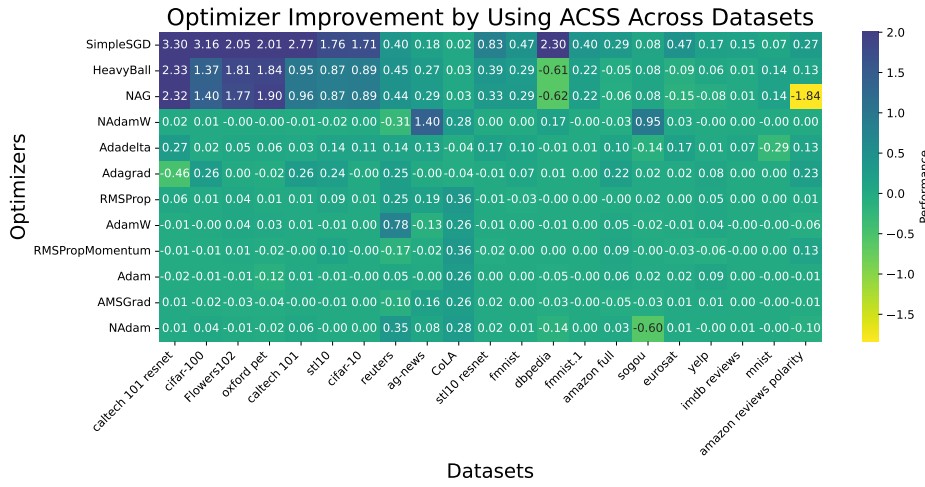

Figure 3: Quantitative improvement in training loss using ACSS across datasets and optimizers after a fixed number of epochs.

Table 2: Training Loss over 5 Epochs for Yelp Reviews Polarity Dataset (560,000 reviews) using a Simplified RNN Model. The model consists of embedding, RNN, and fully connected layers. ACSS versions of optimizers generally outperform their traditional counterparts.

| Optimizer Name | Regular Optimizer | | | | | ACSS Version of Optimizer | | | | |
|---|---|---|---|---|---|---|---|---|---|---|
| | Epoch 1 | Epoch 2 | Epoch 3 | Epoch 4 | Epoch 5 | Epoch 1 | Epoch 2 | Epoch 3 | Epoch 4 | Epoch 5 |
| Adadelta | 0.680 ±0.00 | 0.674 ±0.00 | 0.671 ±0.00 | 0.669 ±0.00 | 0.668 ±0.00 | 0.679 ±0.01 | 0.670 ±0.00 | 0.666 ±0.00 | 0.663 ±0.00 | 0.659 ±0.00 |
| Adagrad | 0.558 ±0.01 | 0.521 ±0.01 | 0.510 ±0.01 | 0.501 ±0.01 | 0.493 ±0.01 | 0.569 ±0.07 | 0.498 ±0.07 | 0.452 ±0.06 | 0.429 ±0.06 | 0.410 ±0.07 |
| Adam | 0.627 ±0.01 | 0.584 ±0.01 | 0.587 ±0.00 | 0.568 ±0.04 | 0.575 ±0.02 | 0.542 ±0.04 | 0.541 ±0.16 | 0.530 ±0.17 | 0.457 ±0.20 | 0.489 ±0.14 |
| AdamW | 0.581 ±0.02 | 0.567 ±0.03 | 0.478 ±0.01 | 0.499 ±0.08 | 0.419 ±0.11 | 0.599 ±0.04 | 0.589 ±0.12 | 0.555 ±0.10 | 0.413 ±0.05 | 0.376 ±0.12 |
| AMSGrad | 0.537 ±0.00 | 0.548 ±0.01 | 0.569 ±0.11 | 0.481 ±0.05 | 0.589 ±0.07 | 0.616 ±0.04 | 0.596 ±0.02 | 0.625 ±0.08 | 0.625 ±0.04 | 0.578 ±0.03 |
| HeavyBall | 0.666 ±0.00 | 0.652 ±0.00 | 0.604 ±0.01 | 0.529 ±0.01 | 0.512 ±0.01 | 0.572 ±0.01 | 0.517 ±0.01 | 0.491 ±0.01 | 0.474 ±0.01 | 0.455 ±0.01 |
| NAdam | 0.637 ±0.01 | 0.612 ±0.00 | 0.589 ±0.00 | 0.580 ±0.04 | 0.537 ±0.09 | 0.609 ±0.02 | 0.543 ±0.05 | 0.543 ±0.01 | 0.531 ±0.04 | 0.538 ±0.02 |
| NAdamW | 0.601 ±0.01 | 0.531 ±0.00 | 0.495 ±0.05 | 0.498 ±0.05 | 0.523 ±0.03 | 0.632 ±0.00 | 0.594 ±0.02 | 0.585 ±0.02 | 0.541 ±0.04 | 0.528 ±0.02 |
| NAG | 0.666 ±0.00 | 0.652 ±0.00 | 0.604 ±0.01 | 0.529 ±0.01 | 0.510 ±0.02 | 0.630 ±0.02 | 0.616 ±0.00 | 0.604 ±0.01 | 0.604 ±0.03 | 0.591 ±0.02 |
| RMSProp | 0.650 ±0.04 | 0.538 ±0.07 | 0.495 ±0.13 | 0.425 ±0.09 | 0.447 ±0.03 | 0.624 ±0.02 | 0.493 ±0.03 | 0.432 ±0.03 | 0.407 ±0.06 | 0.394 ±0.06 |
| RMSPropMomentum | 0.652 ±0.02 | 0.578 ±0.04 | 0.561 ±0.03 | 0.491 ±0.05 | 0.467 ±0.03 | 0.633 ±0.06 | 0.601 ±0.03 | 0.581 ±0.00 | 0.551 ±0.04 | 0.524 ±0.04 |
| SimpleSGD | 0.676 ±0.00 | 0.671 ±0.00 | 0.669 ±0.00 | 0.667 ±0.00 | 0.665 ±0.00 | 0.596 ±0.01 | 0.535 ±0.01 | 0.519 ±0.01 | 0.506 ±0.01 | 0.493 ±0.02 |

## 4.2 PERFORMANCE ON THE YELP REVIEWS DATASET

We evaluated various optimizers with and without ACSS on the Yelp Reviews Polarity Dataset (560,000 reviews) using a simplified RNN model. The ACSS variants generally outperformed their standard counterparts over five epochs. AdamW-ACSS showed the most significant improvement, with loss decreasing from 0.5994 to 0.3756 across epochs, outperforming the traditional AdamW's final loss. SimpleSGD-ACSS demonstrated remarkable improvement, matching top performers like AdamW-ACSS by the first epoch.

**Key Takeaways:** The best performing non-ACSS optimizer after Epoch 5 reaches a training loss of only 0.419 (AdamW), which is reached at Epoch 4 for two of the ACSS versions. All the best-performing optimizers after Epoch 2 are ACSS versions of the optimizers.

## 4.3 TRAINING LOSS IMPROVEMENTS AVERAGED OVER ALL DATASETS

We evaluated the performance of Adaptive Curvature Step Size (ACSS) variants of SimpleSGD, HeavyBall, and NAG (Nesterov Accelerated Gradient) across diverse datasets in vision and language domains. Our evaluation encompassed various model architectures, including CNNs (such as ResNet), RNNs, and simple neural networks. The results, as illustrated in Figure 4, demonstrate consistent improvements in training performance for ACSS variants compared to their standard counterparts. These improvements were observed across all five epochs and increased over time, indicating that ACSS provides sustained benefits throughout the training process.

**Key Takeaways:** Optimizers that do not store square-gradient terms (SGD, HeavyBall, NAG) exhibit significant outperformance through the use of ACSS. The improvement in mean training loss, averaged across all datasets, is evident across all the epochs.

## 4.4 PERFORMANCE ON VISION BENCHMARKS

Figure 5 presents a heatmap of optimizer rankings across five vision datasets: Caltech101, CIFAR10, Flowers102, MNIST, and STL10. The analysis reveals that Adadelta and RMSProp variants consistently underperform, with ACSS showing minimal impact on their effectiveness. In contrast, Adam, AdamW, and AMSGrad perform well initially, with ACSS offering marginal improvements. Adagrad demonstrates high performance variance across datasets.

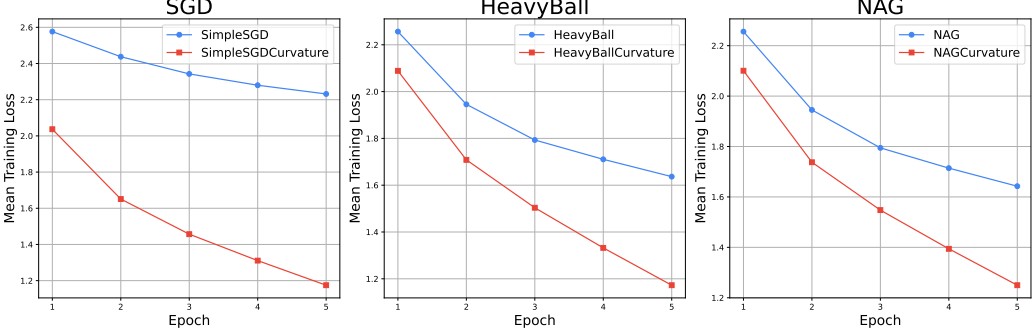

Figure 4: Mean training loss across epochs for different optimizers.

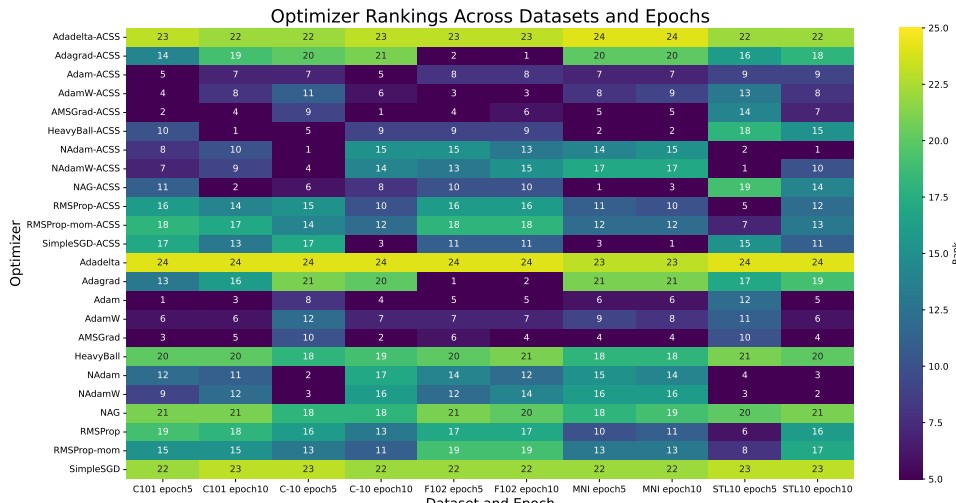

Figure 5: Heatmap of optimizer rankings across various computer vision datasets. The heatmap displays the performance ranks of 24 optimizers, including both standard versions and their Adaptive Curvature Step Size (ACSS) variants, on five different datasets (Caltech101, CIFAR10, Flowers102, MNIST, and STL10) at epochs 5 and 10. Rankings range from 1 (best performing) to 24 (worst performing), with lower numbers and cooler colors indicating better performance. This visualization highlights the impact of ACSS on various optimizers across different datasets.

Notably, optimizers that do not incorporate squared gradients (SimpleSGD, HeavyBall, NAG) benefit most from ACSS. These optimizers achieve performance boosts comparable to methods using squared gradients, but without the associated memory overhead.

**Key Takeaways:** ACSS versions generally outperform their traditional counterparts on these vision benchmarks for both ResNet-18 and simple CNN architectures. The most significant improvements are observed in optimizers that do not initially use squared gradients.

### 4.5 OVERALL RANK IMPROVEMENTS FOR DIFFERENT OPTIMIZERS

Figure 6 illustrates the performance improvement of optimizers with ACSS across multiple datasets. Optimizers with lower memory requirements benefit most from ACSS. SimpleSGD, with the smallest memory footprint, shows the highest average rank improvement of 12.5. HeavyBall and NAG also demonstrate significant enhancements, with average improvements of 7.9 and 6.7 respectively.

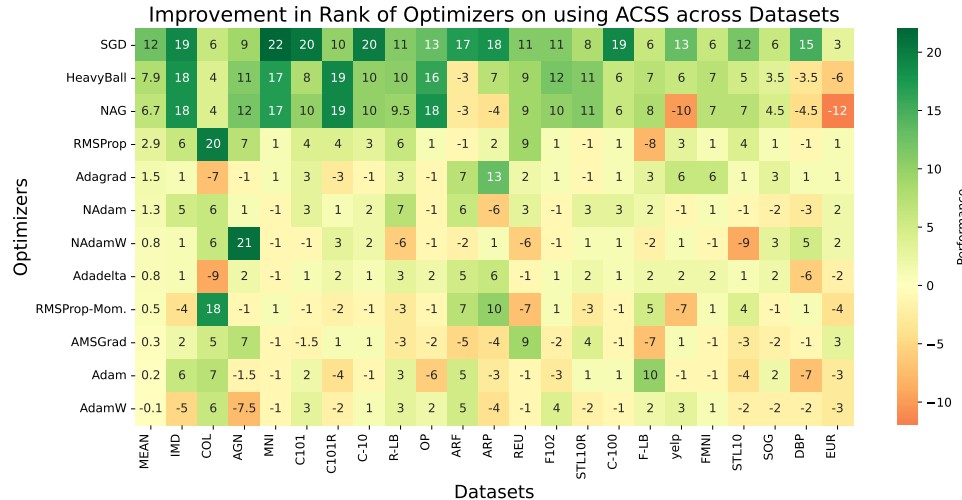

Figure 6: Heatmap of optimizer rank improvements when using ACSS across datasets. Green indicates better performance, red indicates worse. The datasets are listed on the X-axis, and the optimizers on the Y-axis. Color intensity represents the degree of improvement.

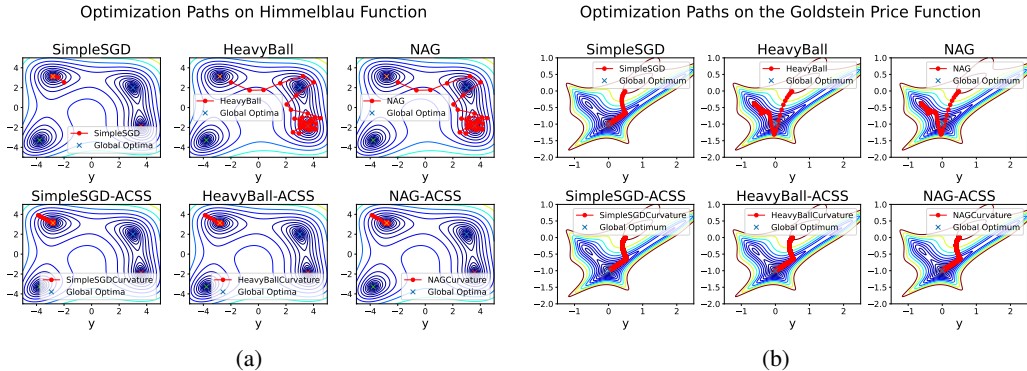

Figure 7: Optimization paths on the Goldstein-Price (left) and Himmelblau (right) functions. These functions present challenges due to their complex landscapes with multiple optima and flat regions. More complex optimizers like Adam, AdamW, and AMSGrad, which already incorporate adaptive learning rate mechanisms, show lower benefits. This suggests ACSS is particularly effective in enhancing simpler optimization algorithms, offering a memory-efficient alternative to more complex adaptive methods.

**Key Takeaways:** Except for AdamW, all optimizers show positive mean performance improvement with ACSS, indicating benefits in incorporating ACSS into existing optimization pipelines.

### 4.6 OPTIMIZATION ON CHALLENGING FUNCTIONS

We now plot the performance of our optimizers on two challenging functions: the Himmelblau and Goldstein-Price functions. Additional functions are analyzed in Appendix F.

**The Himmelblau Function:** The Himmelblau function has four global minima. ACSS versions converge to the nearest minimum from the starting point (-4,4), while other versions overshoot at a learning rate of $1.5 \times 10^{-2}$. At higher rates, non-ACSS versions diverge, whereas ACSS versions maintain convergence.

**The Goldstein-Price Function:** The Goldstein-Price function, with its complex landscape of multiple local minima and one global minimum at (0, -1), challenges gradient-based methods. ACSS optimizers dynamically adjust step sizes based on local curvature, enabling precise convergence to the global minimum. In contrast, standard Heavyball and NAG optimizers overshoot, moving toward different local minima. We plot 5000 iterations from (0.5, 0) with a learning rate of $2.5 \times 10^{-5}$.

**Key Takeaways:** In Figures 1, 7 in the main paper, and Figure 8 in Appendix F, we plot the ACSS performance as compared with the regular versions for challenging optimization benchmark functions. In all the cases, the ACSS versions showed better stability and convergence properties compared to the traditional algorithms.

### 4.7 LIMITATIONS:

It is important to acknowledge that ACSS introduces additional computational overhead per iteration, with theoretical analysis suggesting up to twice the cost and experimental wall-clock time measurements showing an average increase of 1.37 times for the ACSS optimizers over their traditional counterparts, which is balanced against its memory efficiency benefits and lower time to convergence (see Section D for detailed theoretical and experimental analyses).

## 5 CONCLUSIONS

This work introduced the Adaptive Curvature Step Size (ACSS) method, a novel optimization approach that leverages the geometric properties of the optimization path to dynamically adjust step sizes. Our comprehensive empirical evaluation across diverse datasets and challenging functions demonstrates that ACSS consistently outperforms traditional optimization methods. The method's ability to incorporate second-order-like information without explicit computation of the Hessian is a key benefit, as we show through our theoretical guarantees. Furthermore, ACSS's low memory footprint makes it particularly suitable for large-scale optimization setups and low-resource settings. The generalized framework we provide for incorporating ACSS into various optimization algorithms, along with our PyTorch implementations, facilitates further research in this direction.

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
