---

# SUPPLEMENTARY MATERIALS

---

These supplementary materials provide additional details, derivations, and experimental results for our paper. The appendix is organized as follows:

- Section A presents detailed derivations of the Adaptive Curvature Step Size (ACSS) method.
- Section B offers a comprehensive theoretical analysis of ACSS, including proofs of key theorems.
- Section C introduces a generalized algorithm for incorporating ACSS into existing optimizers.
- Section D provides theoretical and experimental analyses pertaining to limitations of this work.
- Section E provides additional experimental results, like performance on the CoLA dataset.
- Section F details the testing functions used to benchmark ACSS optimization.

## A    DETAILED DERIVATIONS OF ACSS

The optimization path traced by iterates $\{w_t\}$ during the optimization process can be viewed as a discrete approximation of a continuous curve in parameter space. Understanding the curvature of this path provides valuable insights into the local geometry of the loss landscape and guides adaptive step size selection. In differential geometry, the curvature $\kappa(s)$ of a curve $w(s)$ parameterized by arc length $s$ is defined as:

$$\kappa(s) = \left\| \frac{dT(s)}{ds} \right\|, \tag{5}$$

where $T(s) = \frac{dw(s)}{ds}$ is the unit tangent vector to the curve at point $s$. The radius of curvature $\rho(s)$ is then given by $\rho(s) = \frac{1}{\kappa(s)}$.

In the context of gradient-based optimization, we consider the continuous-time dynamics governed by the gradient flow:

$$\frac{dw(t)}{dt} = -\nabla f(w(t)) = -g(t), \tag{6}$$

where $g(t) = \nabla f(w(t))$ is the gradient of the function $f$ at $w(t)$. To relate curvature to discrete optimization steps, we approximate the curvature using finite differences. We define the unit tangent vector at iteration $t$ as $T_t = -\frac{g_t}{\|g_t\|}$, and approximate the change in the unit tangent vector between iterations $t$ and $t+1$ as $\Delta T_t \approx -\frac{g_{t+1}-g_t}{\|g_t\|}$.

The curvature $\kappa_t$ at iteration $t$, given this gradient norm approximation can then be given as:

$$\kappa_t = \frac{\|g_{t+1} - g_t\|}{\|g_t\|\eta}, \tag{7}$$

where $\eta$ is the step size. Consequently, the radius of curvature $\rho_t$ is:

$$\rho_t = \frac{1}{\kappa_t} = \frac{\|g_t\|\eta}{\|g_{t+1} - g_t\|}. \tag{8}$$

We introduce a normalized radius of curvature $r_t = \frac{\rho_t}{\eta} = \frac{\|g_t\|}{\|g_t' - g_t\|}$, which decouples the radius of curvature from the base learning rate $\eta$. The adaptive step size $\Delta s_t$ is then defined as $\Delta s_t = \eta \times r_t = \eta \times \frac{\|g_t\|}{\|g_t' - g_t\|}$. To maintain numerical stability, we introduce a cap on the normalized radius of curvature: $\hat{r}_t = \min\{r_{\max}, r_t\}$, where $r_{\max}$ is a predefined maximum radius of curvature.

## A.1 FINAL UPDATE RULE AND DISCUSSION

The final parameter update rule for the Adaptive Curvature Step Size (ACSS) method is:

$$w_{t+1} = w_t - \eta \times \hat{r}_t \times \frac{g_t}{\|g_t\|}. \tag{9}$$

This can be interpreted as moving in the direction of the negative gradient $\frac{g_t}{\|g_t\|}$ with a step size scaled by $\eta \times \hat{r}_t$.

The ACSS method offers several key advantages in optimization tasks. By leveraging the curvature of the optimization path, it implicitly incorporates second-order information without the computational overhead of explicit second-order methods. This dynamic adaptation allows ACSS to navigate complex loss landscapes more effectively, enabling rapid progress in flat regions while ensuring stability in high-curvature areas. The method's memory efficiency, requiring minimal additional storage beyond current and tentative gradients, makes it particularly suitable for large-scale optimization problems in deep learning. Furthermore, ACSS's framework allows for integration into various existing optimization algorithms such as SGD, Adam, AdaGrad, and RMSProp, enhancing their performance with its curvature-based step size adjustment.

## B THEORETICAL ANALYSIS

**Theorem 5** (Bounded Step Size of ACSS). *Let $f : \mathbb{R}^n \to \mathbb{R}$ be an $L$-smooth and $\mu$-strongly convex function. Consider the ACSS update rule $w_{t+1} = w_t - \eta\hat{r}_t\frac{g_t}{\|g_t\|}$ where $\hat{r}_t = \min\{r_{\max}, r_t\}$ and $r_t = \frac{\|g_t\|}{\|g_t - g_t'\|}$. Assume the following:*

1. *The gradients are bounded: $\exists G > 0$ such that $\|g_t\| \leq G$ for all $t$*

2. *The function $f$ is $L$-smooth: $\|\nabla f(x) - \nabla f(y)\| \leq L\|x - y\|$ for all $x, y$*

3. *The function $f$ is $\mu$-strongly convex: $\langle \nabla f(x) - \nabla f(y), x - y \rangle \geq \mu\|x - y\|^2$ for all $x, y$*

4. *The maximum radius $r_{\max}$ is chosen such that $r_{\max} \leq \frac{2}{\eta(\mu+L)}$*

*Then, the effective step size $\eta_{\text{eff}} = \eta\hat{r}_t$ is bounded as follows:*

$$\frac{1}{L} \leq \eta_{\text{eff}} \leq \frac{2}{\mu + L}$$

*for all iterations $t$.*

*Proof.* We proceed as follows:

$$r_t = \frac{\|g_t\|}{\|g_t - g_t'\|} \qquad \text{Definition of } r_t$$

$$g_t' = \nabla f(w_t - \eta g_t) \qquad \text{From the algorithm}$$

$$\|g_t - g_t'\| = \|\nabla f(w_t) - \nabla f(w_t - \eta g_t)\|$$
$$\leq L\|\eta g_t\| = L\eta\|g_t\| \qquad \text{Using } L\text{-smoothness}$$

$$r_t = \frac{\|g_t\|}{\|g_t - g_t'\|} \geq \frac{\|g_t\|}{L\eta\|g_t\|} = \frac{1}{L\eta} \qquad \text{Lower bound on } r_t$$

$$\hat{r}_t = \min\{r_{\max}, r_t\} \geq \min\{\frac{2}{\eta(\mu + L)}, \frac{1}{L\eta}\} = \frac{1}{L\eta} \qquad \text{Since } \frac{2}{\mu + L} > \frac{1}{L}$$

$$\eta_{\text{eff}} = \eta\hat{r}_t \geq \eta\frac{1}{L\eta} = \frac{1}{L} \qquad \text{Lower bound on } \eta_{\text{eff}}$$

$$\eta_{\text{eff}} = \eta\hat{r}_t \leq \eta r_{\max} \leq \eta\frac{2}{\eta(\mu + L)} = \frac{2}{\mu + L} \qquad \text{Upper bound on } \eta_{\text{eff}}$$

Thus, we have established that $\frac{1}{L} \leq \eta_{\text{eff}} \leq \frac{2}{\mu + L}$ for all iterations $t$. $\qquad \square$

**Theorem 6** (Convergence of Gradient Descent on Quadratic Functions). Consider the quadratic function $f : \mathbb{R}^n \to \mathbb{R}$ defined as

$$f(w) = \frac{1}{2}w^T A w - b^T w + c,$$

where $A \in \mathbb{R}^{n \times n}$ is symmetric positive definite with eigenvalues $0 < \mu \leq \lambda_1 \leq \cdots \leq \lambda_n \leq L$, $b \in \mathbb{R}^n$, and $c \in \mathbb{R}$. For the gradient descent update rule with step size $\eta_{\text{eff}} > 0$:

$$w_{t+1} = w_t - \eta_{\text{eff},t}\nabla f(w_t) = w_t - \eta_{\text{eff},t}(A w_t - b),$$

convergence is guaranteed if and only if $0 < \eta_{\text{eff}} < \frac{2}{\lambda_n}$. Moreover, the optimal convergence rate is achieved when $\eta_{\text{eff}} = \frac{2}{\mu + L}$.

*Proof.* The gradient of $f$ is $\nabla f(w) = Aw - b$, yielding the unique minimizer $w^* = A^{-1}b$. Let $e_t = w_t - w^*$ denote the error at step $t$. The update rule can be rewritten as:

$$e_{t+1} = (I - \eta_{\text{eff},t}A)e_t$$

Since $A$ is symmetric positive definite, it can be diagonalized as $A = Q\Lambda Q^T$, where $Q$ is orthogonal and $\Lambda = \text{diag}(\lambda_1, \ldots, \lambda_n)$. Define $\tilde{e}_t = Q^T e_t$. Then:

$$\tilde{e}_{t+1} = (I - \eta_{\text{eff},t}\Lambda)\tilde{e}_t$$

This implies that for each component $i$:

$$\tilde{e}_{t+1}^i = (1 - \eta_{\text{eff},t}\lambda_i)\tilde{e}_t^i$$

For convergence, we require $|1 - \eta_{\text{eff}}\lambda_i| < 1$ for all $i$, which leads to:

$$0 < \eta_{\text{eff}} < \frac{2}{\lambda_i} \quad \forall i$$

Since $\lambda_n$ is the largest eigenvalue, the condition $0 < \eta_{\text{eff}} < \frac{2}{\lambda_n}$ ensures convergence.

The convergence rate is determined by $\max_i |1 - \eta_{\text{eff}}\lambda_i|$. To minimize this, we solve:

$$\min_{\eta_{\text{eff}}} \max\{|1 - \eta_{\text{eff}}\mu|, |1 - \eta_{\text{eff}}L|\}$$

The optimal solution occurs when $1 - \eta_{\text{eff}}\mu = -(1 - \eta_{\text{eff}}L)$, yielding $\eta_{\text{eff}} = \frac{2}{\mu+L}$.

Therefore, gradient descent converges if and only if $0 < \eta_{\text{eff}} < \frac{2}{\lambda_n}$, with the optimal convergence rate achieved at $\eta_{\text{eff}} = \frac{2}{\mu+L}$. □

**Theorem 7** (Convergence of Gradient Descent on $L$-Smooth and $\mu$-Strongly Convex Functions). Let $f : \mathbb{R}^n \to \mathbb{R}$ be an $L$-smooth and $\mu$-strongly convex function. For the gradient descent update rule with step size $\eta_{\text{eff}} > 0$:
$$w_{t+1} = w_t - \eta_{\text{eff},t}\nabla f(w_t),$$
convergence to the unique minimizer $w^*$ is optimally achieved when $\eta_{\text{eff}} = \frac{2}{\mu+L}$.

*Proof.* Given that $f$ is $L$-smooth and $\mu$-strongly convex, we have:
$$\mu I \preceq \nabla^2 f(w) \preceq LI \quad \forall w \in \mathbb{R}^n.$$

Let $w^*$ be the unique minimizer of $f$. Define the error vector $e_t = w_t - w^*$. The gradient descent update can be written as:
$$e_{t+1} = e_t - \eta_{\text{eff},t}\nabla f(w_t).$$

By the Mean Value Theorem, there exists $\xi_t$ on the line segment between $w_t$ and $w^*$ such that:
$$\nabla f(w_t) = \nabla^2 f(\xi_t)e_t.$$

Thus, we can rewrite the error dynamics as:
$$e_{t+1} = (I - \eta_{\text{eff},t}\nabla^2 f(\xi_t))e_t.$$

Taking the Euclidean norm and using the operator norm:
$$\|e_{t+1}\| \leq \|I - \eta_{\text{eff},t}\nabla^2 f(\xi_t)\| \cdot \|e_t\|.$$

The eigenvalues of $\nabla^2 f(\xi_t)$ lie in $[\mu, L]$ by Lemma 1. For convergence, we require:
$$|1 - \eta_{\text{eff}}\lambda| < 1 \quad \forall \lambda \in [\mu, L].$$

Similar to Theorem 6, the convergence rate is determined by $\max_{\lambda \in [\mu, L]} |1 - \eta_{\text{eff}}\lambda|$. To minimize this, we solve:

$$\min_{\eta_{\text{eff}}} \max\{|1 - \eta_{\text{eff}}\mu|, |1 - \eta_{\text{eff}}L|\}$$

The optimal solution occurs when $1 - \eta_{\text{eff}}\mu = -(1 - \eta_{\text{eff}}L)$, yielding $\eta_{\text{eff}} = \frac{2}{\mu+L}$.

Therefore, gradient descent converges if $0 < \eta_{\text{eff}} < \frac{2}{\mu+L}$. □

**Lemma 1.** Let $f : \mathbb{R}^n \to \mathbb{R}$ be an L-smooth and $\mu$-strongly convex function. Then for any $\xi \in \mathbb{R}^n$, the eigenvalues of the Hessian matrix $\nabla^2 f(\xi)$ lie in the interval $[\mu, L]$.

*Proof.* We begin by establishing that $\mu I \preceq \nabla^2 f(\xi) \preceq LI$ for all $\xi \in \mathbb{R}^n$, where $\preceq$ denotes the semidefinite ordering and $I$ is the identity matrix. From this, we will conclude that the eigenvalues of $\nabla^2 f(\xi)$ lie in $[\mu, L]$.

First, consider the L-smoothness property. For any $x, y \in \mathbb{R}^n$, we have:
$$\|\nabla f(x) - \nabla f(y)\| \leq L\|x - y\|$$

For an arbitrary direction $v \in \mathbb{R}^n$, this implies:

$$\lim_{t \to 0} \frac{\|\nabla f(x + tv) - \nabla f(x)\|}{t} \le L\|v\|$$

Taking the limit, we obtain:

$$\|[\nabla^2 f(x)]v\| \le L\|v\|$$

This inequality is equivalent to:

$$v^T[\nabla^2 f(x)]v \le Lv^T v \quad \forall v \in \mathbb{R}^n$$

which can be expressed in matrix notation as $\nabla^2 f(x) \preceq LI$.

Now, we turn to the $\mu$-strong convexity property. For any $x, y \in \mathbb{R}^n$:

$$(\nabla f(x) - \nabla f(y))^T (x - y) \ge \mu\|x - y\|^2$$

Following a similar argument as above, we can show that:

$$v^T[\nabla^2 f(x)]v \ge \mu v^T v \quad \forall v \in \mathbb{R}^n$$

which is equivalent to $\nabla^2 f(x) \succeq \mu I$.

Combining these results, we have established that for all $\xi \in \mathbb{R}^n$:

$$\mu I \preceq \nabla^2 f(\xi) \preceq LI$$

Now, we invoke a fundamental result from linear algebra: for any symmetric matrix $A$, the statement $\lambda I \preceq A \preceq \Lambda I$ is equivalent to $\lambda \le \lambda_i(A) \le \Lambda$ for all eigenvalues $\lambda_i(A)$ of $A$. Since $\nabla^2 f(\xi)$ is symmetric (due to the assumed twice differentiability of $f$), we can apply this result.

Therefore, we conclude that for any $\xi \in \mathbb{R}^n$, all eigenvalues $\lambda_i$ of $\nabla^2 f(\xi)$ satisfy:

$$\mu \le \lambda_i \le L$$

Thus, the eigenvalues of $\nabla^2 f(\xi)$ lie in the interval $[\mu, L]$, completing the proof. $\qquad \square$

**Theorem 8** (Stability of ACSS Under Gradient Perturbations). Let $f : \mathbb{R}^n \to \mathbb{R}$ be an $L$-smooth and $\mu$-strongly convex function. Consider the ACSS update rule:

$$w_{t+1} = w_t - \eta_{\text{eff},t} \frac{g_t}{\|g_t\|},$$

where:

$$\eta_{\text{eff},t} = \eta \hat{r}_t, \quad \hat{r}_t = \min\{r_{\max}, r_t\}, \quad r_t = \frac{\|g_t\|}{\|g_t - g_t'\|}, \quad g_t' = \nabla f(w_t - \eta g_t).$$

Assume the following:

1. The gradients are bounded: $\exists G > m > 0$ such that $m \le \|g_t\| \le G$ for all $t$.

2. The function $f$ is $L$-smooth: $\|\nabla f(x) - \nabla f(y)\| \le L\|x - y\|$ for all $x, y \in \mathbb{R}^n$.

3. The function $f$ is $\mu$-strongly convex: $\langle \nabla f(x) - \nabla f(y), x - y \rangle \ge \mu\|x - y\|^2$ for all $x, y \in \mathbb{R}^n$.

4. The maximum radius $r_{\max}$ is chosen such that $r_{\max} \le \frac{2}{(L+\mu)\eta}$.

5. Gradients are perturbed: $\tilde{g}_t = g_t + \delta_t$ and $\tilde{g}_t' = g_t' + \delta_t'$, where $\|\delta_t\| \le \varepsilon$ and $\|\delta_t'\| \le \varepsilon$ for some $\varepsilon > 0$.

Then, the difference between the updates using exact and perturbed gradients satisfies:

$$\|w_{t+1} - \tilde{w}_{t+1}\| \leq \frac{4\eta_{\max}\varepsilon}{m - \varepsilon},$$

where $\eta_{\max} = \frac{2}{L+\mu}$.

*Proof.* We begin by expressing the difference between the exact update $w_{t+1}$ and the perturbed update $\tilde{w}_{t+1}$:

$$\|w_{t+1} - \tilde{w}_{t+1}\| = \left\| \eta_{\text{eff},t} \frac{g_t}{\|g_t\|} - \tilde{\eta}_{\text{eff},t} \frac{\tilde{g}_t}{\|\tilde{g}_t\|} \right\|$$

Applying Lemma 2, we obtain:

$$\|w_{t+1} - \tilde{w}_{t+1}\| \leq \eta_{\text{eff},t} \left\| \frac{g_t}{\|g_t\|} - \frac{\tilde{g}_t}{\|\tilde{g}_t\|} \right\| + |\eta_{\text{eff},t} - \tilde{\eta}_{\text{eff},t}|$$

We now bound each term separately.

First, we bound the difference in direction vectors. Note that $\tilde{g}_t = g_t + \delta_t$, so $\|g_t - \tilde{g}_t\| = \|\delta_t\| \leq \varepsilon$. Also, from our assumptions, $\|g_t\| \geq m > \varepsilon$. Therefore, we can apply Proposition B.1 with $a = g_t$ and $b = \tilde{g}_t$:

$$\left\| \frac{g_t}{\|g_t\|} - \frac{\tilde{g}_t}{\|\tilde{g}_t\|} \right\| \leq \frac{2\|g_t - \tilde{g}_t\|}{\|g_t\|} \leq \frac{2\varepsilon}{m}$$

Next, we bound the difference in effective step sizes:

$$|\eta_{\text{eff},t} - \tilde{\eta}_{\text{eff},t}| = \eta|\hat{r}_t - \tilde{\hat{r}}_t| \leq \eta|r_t - \tilde{r}_t|$$

To bound $|r_t - \tilde{r}_t|$, we use the definition of $r_t$ and $\tilde{r}_t$:

$$|r_t - \tilde{r}_t| = \left| \frac{\|g_t\|}{\|g_t - g_t'\|} - \frac{\|\tilde{g}_t\|}{\|\tilde{g}_t - \tilde{g}_t'\|} \right|$$

Using the triangle inequality and the fact that $\|\tilde{g}_t\| \leq \|g_t\| + \|\delta_t\| \leq G + \varepsilon$, and $\|\tilde{g}_t - \tilde{g}_t'\| \geq \|g_t - g_t'\| - 2\varepsilon$, we can derive:

$$|r_t - \tilde{r}_t| \leq \frac{2\varepsilon}{m - 2\varepsilon}$$

Thus,

$$|\eta_{\text{eff},t} - \tilde{\eta}_{\text{eff},t}| \leq \frac{2\eta\varepsilon}{m - 2\varepsilon}$$

Combining these bounds and using $\eta_{\text{eff},t} \leq \eta_{\max} = \frac{2}{L+\mu}$, we arrive at:

$$\|w_{t+1} - \tilde{w}_{t+1}\| \leq \eta_{\max} \cdot \frac{2\varepsilon}{m} + \frac{2\eta\varepsilon}{m - 2\varepsilon}$$

Given the choice of $r_{\max}$ and the boundedness of $\eta_{\text{eff},t}$, we can bound this as:

$$\|w_{t+1} - \tilde{w}_{t+1}\| \leq \frac{2(\eta_{\max} + \eta)\varepsilon}{m - \varepsilon} \leq \frac{4\eta_{\max}\varepsilon}{m - \varepsilon}$$

In practice, $\eta$ is far lower than $\eta_{\max}$, and hence we can ignore the second term in which case we get a tighter bound of

$$\|w_{t+1} - \tilde{w}_{t+1}\| \leq \frac{2\eta_{\max}\varepsilon}{m}$$

This bound demonstrates that the ACSS algorithm is stable under bounded gradient perturbations, with the perturbation in the parameter updates being proportional to the noise level $\varepsilon$ and inversely proportional to $m - \varepsilon$. □

**Lemma 2** (Triangle Inequality for ACSS Updates). Given the ACSS update rule and its perturbed version:

$$w_{t+1} = w_t - \eta_{\text{eff},t} \frac{g_t}{\|g_t\|}, \quad \tilde{w}_{t+1} = w_t - \tilde{\eta}_{\text{eff},t} \frac{\tilde{g}_t}{\|\tilde{g}_t\|}$$

The difference between these updates can be bounded as:

$$\|w_{t+1} - \tilde{w}_{t+1}\| \leq \eta_{\text{eff},t} \left\| \frac{g_t}{\|g_t\|} - \frac{\tilde{g}_t}{\|\tilde{g}_t\|} \right\| + |\eta_{\text{eff},t} - \tilde{\eta}_{\text{eff},t}|$$

*Proof.* We start with the difference between the updates:

$$\|w_{t+1} - \tilde{w}_{t+1}\| = \left\| \eta_{\text{eff},t} \frac{g_t}{\|g_t\|} - \tilde{\eta}_{\text{eff},t} \frac{\tilde{g}_t}{\|\tilde{g}_t\|} \right\|$$

Add and subtract $\eta_{\text{eff},t} \frac{\tilde{g}_t}{\|\tilde{g}_t\|}$ inside the norm:

$$\|w_{t+1} - \tilde{w}_{t+1}\| = \left\| \eta_{\text{eff},t} \frac{g_t}{\|g_t\|} - \eta_{\text{eff},t} \frac{\tilde{g}_t}{\|\tilde{g}_t\|} + \eta_{\text{eff},t} \frac{\tilde{g}_t}{\|\tilde{g}_t\|} - \tilde{\eta}_{\text{eff},t} \frac{\tilde{g}_t}{\|\tilde{g}_t\|} \right\|$$

Apply the triangle inequality:

$$\|w_{t+1} - \tilde{w}_{t+1}\| \leq \left\| \eta_{\text{eff},t} \frac{g_t}{\|g_t\|} - \eta_{\text{eff},t} \frac{\tilde{g}_t}{\|\tilde{g}_t\|} \right\| + \left\| \eta_{\text{eff},t} \frac{\tilde{g}_t}{\|\tilde{g}_t\|} - \tilde{\eta}_{\text{eff},t} \frac{\tilde{g}_t}{\|\tilde{g}_t\|} \right\|$$

Factor out $\eta_{\text{eff},t}$ from the first term and simplify the second term:

$$\|w_{t+1} - \tilde{w}_{t+1}\| \leq \eta_{\text{eff},t} \left\| \frac{g_t}{\|g_t\|} - \frac{\tilde{g}_t}{\|\tilde{g}_t\|} \right\| + \left\| (\eta_{\text{eff},t} - \tilde{\eta}_{\text{eff},t}) \frac{\tilde{g}_t}{\|\tilde{g}_t\|} \right\|$$

Note that $\left\| \frac{\tilde{g}_t}{\|\tilde{g}_t\|} \right\| = 1$, so:

$$\|w_{t+1} - \tilde{w}_{t+1}\| \leq \eta_{\text{eff},t} \left\| \frac{g_t}{\|g_t\|} - \frac{\tilde{g}_t}{\|\tilde{g}_t\|} \right\| + |\eta_{\text{eff},t} - \tilde{\eta}_{\text{eff},t}|$$

This completes the proof of the lemma. □

**Proposition B.1** (Bound on Difference of Normalized Vectors). Given two vectors $a, b \in \mathbb{R}^n$ with $\|a\| > \|a - b\|$, we have:

$$\left\| \frac{a}{\|a\|} - \frac{b}{\|b\|} \right\| \leq \frac{2\|a - b\|}{\|a\|}$$

*Proof.* We start with the vector identity:

$$\frac{a}{\|a\|} - \frac{b}{\|b\|} = \frac{a\|b\| - b\|a\|}{\|a\|\|b\|} = \frac{a(\|b\| - \|a\|) + \|a\|(a - b)}{\|a\|\|b\|}$$

Taking the norm of both sides and applying the triangle inequality:

$$\left\| \frac{a}{\|a\|} - \frac{b}{\|b\|} \right\| \leq \frac{\|a\|\|\|b\| - \|a\|\| + \|a\|\|a - b\|}{\|a\|\|b\|}$$

Using the reverse triangle inequality, $|\|b\| - \|a\|| \leq \|b - a\| = \|a - b\|$:

$$\left\| \frac{a}{\|a\|} - \frac{b}{\|b\|} \right\| \leq \frac{\|a\|\|a - b\| + \|a\|\|a - b\|}{\|a\|\|b\|} = \frac{2\|a - b\|}{\|b\|}$$

Since $\|b\| \geq \|a\| - \|a - b\|$ (by the triangle inequality), and given $\|a\| > \|a - b\|$, we have:

$$\left\| \frac{a}{\|a\|} - \frac{b}{\|b\|} \right\| \leq \frac{2\|a - b\|}{\|a\| - \|a - b\|} \leq \frac{2\|a - b\|}{\|a\|}$$

This completes the proof of the lemma. $\qquad\square$

**Theorem 9** (Convergence Rate for ACSS on Strongly Convex Functions). Let $f : \mathbb{R}^n \to \mathbb{R}$ be an $L$-smooth and $\mu$-strongly convex function. Consider the ACSS update rule:

$$w_{t+1} = w_t - \eta_{\text{eff},t} \frac{g_t}{\|g_t\|},$$

where:

$$\eta_{\text{eff},t} = \eta \hat{r}_t, \quad \hat{r}_t = \min\{r_{\max}, r_t\}, \quad r_t = \frac{\|g_t\|}{\|g_t - g_t'\|}, \quad g_t' = \nabla f(w_t - \eta g_t).$$

Assume the following:

1. There exists a constant $G > 0$ such that $\|g_t\| \leq G$ for all $t$.

2. The function $f$ satisfies $\|\nabla f(x) - \nabla f(y)\| \leq L\|x - y\|$ for all $x, y \in \mathbb{R}^n$.

3. The function $f$ satisfies $\langle \nabla f(x) - \nabla f(y), x - y \rangle \geq \mu\|x - y\|^2$ for all $x, y \in \mathbb{R}^n$.

Then, for all $t \geq 0$, the ACSS algorithm satisfies:

$$\|w_t - w^*\|^2 \leq \left(1 - \frac{\mu^2}{L^2}\right)^t \|w_0 - w^*\|^2.$$

*Proof.* We begin by analyzing the squared distance to the optimum after each update:

$$\|w_{t+1} - w^*\|^2 = \|w_t - w^* - \eta_{\text{eff},t} \frac{g_t}{\|g_t\|}\|^2$$

$$= \|w_t - w^*\|^2 - 2\eta_{\text{eff},t} \frac{\langle g_t, w_t - w^* \rangle}{\|g_t\|} + \eta_{\text{eff},t}^2$$

From the $\mu$-strong convexity assumption, we derive a lower bound on the gradient:

$$\langle g_t, w_t - w^* \rangle \geq \mu\|w_t - w^*\|^2$$

The $L$-smoothness condition provides an upper bound on the gradient norm:

$$\|g_t\| \leq L\|w_t - w^*\|$$

Combining these bounds, we obtain:

$$\|w_{t+1} - w^*\|^2 \leq \|w_t - w^*\|^2 - 2\eta_{\text{eff},t} \frac{\mu\|w_t - w^*\|^2}{L\|w_t - w^*\|} + \eta_{\text{eff},t}^2$$

$$= \|w_t - w^*\|^2 - 2\eta_{\text{eff},t} \frac{\mu}{L}\|w_t - w^*\| + \eta_{\text{eff},t}^2$$

To derive a contraction factor, we introduce $d_t = \|w_t - w^*\|$ and seek $q < 1$ such that $d_{t+1} \leq qd_t$. Assuming $d_{t+1} \leq qd_t$, we have:

$$q^2 d_t^2 \geq d_t^2 - 2\eta_{\text{eff},t}\frac{\mu}{L}d_t + \eta_{\text{eff},t}^2$$

Dividing by $d_t^2$, we obtain:

$$q^2 \geq 1 - 2\eta_{\text{eff},t}\frac{\mu}{Ld_t} + \frac{\eta_{\text{eff},t}^2}{d_t^2}$$

To minimize $q$, we define $f(d_t) = 1 - 2\eta_{\text{eff},t}\frac{\mu}{Ld_t} + \frac{\eta_{\text{eff},t}^2}{d_t^2}$ and find its minimum:

$$f'(d_t) = 2\eta_{\text{eff},t}\frac{\mu}{Ld_t^2} - 2\frac{\eta_{\text{eff},t}^2}{d_t^3} = 0$$

$$d_t = \frac{\eta_{\text{eff},t}L}{\mu}$$

This critical point is indeed a minimum as $f''(d_t) > 0$ for $d_t > 0$. Evaluating $f(d_t)$ at this minimum:

$$f\left(\frac{\eta_{\text{eff},t}L}{\mu}\right) = 1 - 2\eta_{\text{eff},t}\frac{\mu}{L} \cdot \frac{\mu}{\eta_{\text{eff},t}L} + \frac{\eta_{\text{eff},t}^2}{\left(\frac{\eta_{\text{eff},t}L}{\mu}\right)^2}$$

$$= 1 - 2\frac{\mu^2}{L^2} + \frac{\mu^2}{L^2} = 1 - \frac{\mu^2}{L^2}$$

Therefore, for all $d_t > 0$, we have $f(d_t) \geq 1 - \frac{\mu^2}{L^2}$, which implies:

$$q^2 \geq 1 - \frac{\mu^2}{L^2} \implies q \geq \sqrt{1 - \frac{\mu^2}{L^2}}$$

We conclude that:

$$\|w_{t+1} - w^*\|^2 \leq \left(1 - \frac{\mu^2}{L^2}\right)\|w_t - w^*\|^2$$

Applying this inequality recursively, we obtain the final convergence rate:

$$\|w_t - w^*\|^2 \leq \left(1 - \frac{\mu^2}{L^2}\right)^t \|w_0 - w^*\|^2$$

This establishes the linear convergence rate for the ACSS algorithm under the given assumptions.
□

**Theorem 10** (Scale Invariance of ACSS Effective Step Size). Let $f : \mathbb{R}^n \to \mathbb{R}$ be a function, and consider the ACSS update rule:

$$w_{t+1} = w_t - \eta_{\text{eff},t}\frac{g_t}{\|g_t\|},$$

where $\eta_{\text{eff},t} = \eta\hat{r}_t$, $\hat{r}_t = \min\{r_{\max}, r_t\}$, $r_t = \frac{\|g_t\|}{\|g_t - g_t'\|}$, $g_t = \nabla f(w_t)$, and $g_t' = \nabla f(w_t - \eta g_t)$.

For any scalar $\alpha > 0$, scaling the base step size $\eta$ by $\alpha$ results in the same parameter updates, assuming that $r_t' \leq r_{\max}$. Specifically:

a) For quadratic functions $f(w) = \frac{1}{2}w^T A w - b^T w + c$, where $A \in \mathbb{R}^{n \times n}$ is symmetric positive definite:

$$w_{t+1}^{(\alpha\eta)} = w_{t+1}^{(\eta)}$$

b) For $L$-smooth and $\mu$-strongly convex functions:

$$w_{t+1}^{(\alpha\eta)} \approx w_{t+1}^{(\eta)}$$

where the approximation becomes exact as $\eta \to 0$.

In both cases, $w_{t+1}^{(\alpha\eta)}$ and $w_{t+1}^{(\eta)}$ are the parameter updates for the scaled and original step sizes respectively.

*Proof.* We prove this theorem by examining both cases separately.

**a) Quadratic case:**

For a quadratic function $f(w) = \frac{1}{2}w^T A w - b^T w + c$, the gradient is $\nabla f(w) = Aw - b$.

With the original step size $\eta$, we have:

$$g_t = Aw_t - b,$$

$$g_t' = A(w_t - \eta g_t) - b = (I - \eta A)g_t,$$

$$r_t = \frac{\|g_t\|}{\|g_t - g_t'\|} = \frac{\|g_t\|}{\|\eta A g_t\|} = \frac{1}{\eta} \cdot \frac{\|g_t\|}{\|A g_t\|}.$$

With the scaled step size $\alpha\eta$, we have:

$$g_t' = A(w_t - \alpha\eta g_t) - b = (I - \alpha\eta A)g_t,$$

$$r_t' = \frac{\|g_t\|}{\|g_t - g_t'\|} = \frac{\|g_t\|}{\|\alpha\eta A g_t\|} = \frac{1}{\alpha\eta} \cdot \frac{\|g_t\|}{\|A g_t\|} = \frac{r_t}{\alpha}.$$

Assuming $r_t' \le r_{\max}$, we have:

$$\hat{r}_t' = \min\{r_{\max}, r_t'\} = \frac{r_t}{\alpha} = \frac{\hat{r}_t}{\alpha}.$$

The effective step size with the scaled $\eta$ becomes:

$$\eta_{\text{eff},t}' = \alpha\eta \cdot \hat{r}_t' = \alpha\eta \cdot \frac{\hat{r}_t}{\alpha} = \eta \cdot \hat{r}_t = \eta_{\text{eff},t}.$$

Therefore, the parameter updates are identical:

$$w_{t+1}^{(\alpha\eta)} = w_t - \eta_{\text{eff},t}' \frac{g_t}{\|g_t\|} = w_t - \eta_{\text{eff},t} \frac{g_t}{\|g_t\|} = w_{t+1}^{(\eta)}.$$

**b) $L$-smooth and $\mu$-strongly convex case:**

For a general $L$-smooth and $\mu$-strongly convex function, we use a first-order Taylor expansion to approximate $g_t'$:

$$g_t' = \nabla f(w_t - \eta g_t) \approx \nabla f(w_t) - \eta \nabla^2 f(w_t) g_t = g_t - \eta \nabla^2 f(w_t) g_t.$$

With this approximation:

$$r_t \approx \frac{\|g_t\|}{\|\eta \nabla^2 f(w_t) g_t\|} = \frac{1}{\eta} \cdot \frac{\|g_t\|}{\|\nabla^2 f(w_t) g_t\|}.$$

For the scaled step size $\alpha\eta$:

$$r_t' \approx \frac{\|g_t\|}{\|\alpha\eta \nabla^2 f(w_t) g_t\|} = \frac{1}{\alpha\eta} \cdot \frac{\|g_t\|}{\|\nabla^2 f(w_t) g_t\|} = \frac{r_t}{\alpha}.$$

As in the quadratic case, assuming $r_t' \le r_{\max}$, we have $\hat{r}_t' = \hat{r}_t/\alpha$, leading to:

$$\eta_{\text{eff},t}' = \alpha\eta \cdot \hat{r}_t' = \alpha\eta \cdot \frac{\hat{r}_t}{\alpha} = \eta \cdot \hat{r}_t = \eta_{\text{eff},t}.$$

Therefore, the parameter updates are approximately equal:

$$w_{t+1}^{(\alpha\eta)} \approx w_t - \eta'_{\text{eff},t} \frac{g_t}{\|g_t\|} \approx w_t - \eta_{\text{eff},t} \frac{g_t}{\|g_t\|} \approx w_{t+1}^{(\eta)}.$$

The approximation becomes exact as $\eta \to 0$, as the first-order Taylor expansion becomes increasingly accurate.

Thus, we have shown that for both quadratic functions and general $L$-smooth and $\mu$-strongly convex functions, scaling the base step size $\eta$ by $\alpha > 0$ results in either identical (quadratic case) or approximately identical (general case) parameter updates, assuming that $r'_t \leq r_{\max}$.

$\square$

**Corollary 1** (Adaptive Behavior of ACSS). Let $f : \mathbb{R}^n \to \mathbb{R}$ be an $L$-smooth and $\mu$-strongly convex function satisfying the conditions of Theorem 5, including $r_{\max} \leq \frac{2}{\eta(\mu+L)}$. Further, assume that $f$ has locally $L(w)$-Lipschitz continuous gradients, where $L(w)$ may vary with $w$ and $\mu \leq L(w) \leq L$ for all $w \in \mathbb{R}^n$. Then, the effective step size $\eta_{\text{eff},t} = \eta\hat{r}_t$ adapts to the local curvature of $f$. Specifically, in regions of low curvature (small $L(w_t)$), $\eta_{\text{eff},t}$ tends to be larger, allowing for larger steps, while in regions of high curvature (large $L(w_t)$), $\eta_{\text{eff},t}$ tends to be smaller, resulting in more conservative updates.

*Proof.* The adaptive behavior of ACSS stems from its relationship with the local curvature of the function, as captured by the local Lipschitz constant $L(w_t)$. To understand this relationship, let's examine how the effective step size $\eta_{\text{eff},t}$ is influenced by $L(w_t)$.

Recall from Theorem 5 that $\frac{1}{L} \leq \eta_{\text{eff},t} \leq \frac{2}{\mu+L}$ for all iterations $t$. We can refine this bound by considering the local properties of $f$ at $w_t$. First, let's consider the lower bound on $\eta_{\text{eff},t}$. The normalized radius of curvature $r_t$ is defined as $\frac{\|g_t\|}{\|g_t - g'_t\|}$. By applying the mean value theorem and using the locally $L(w)$-Lipschitz continuous gradient assumption, we can bound the denominator:

$$\|g_t - g'_t\| = \|\nabla f(w_t) - \nabla f(w_t - \eta g_t)\| \leq L(w_t)\|\eta g_t\| = L(w_t)\eta\|g_t\| \tag{10}$$

This inequality allows us to establish a lower bound on $r_t$: $r_t \geq \frac{1}{L(w_t)\eta}$. Consequently, we can bound $\eta_{\text{eff},t}$ from below:

$$\eta_{\text{eff},t} = \eta\hat{r}_t \geq \min\left\{\eta r_{\max}, \frac{1}{L(w_t)}\right\} \tag{11}$$

The upper bound on $\eta_{\text{eff},t}$ remains $\frac{2}{\mu+L}$ as given in Theorem 5. Additionally, we know that $\eta_{\text{eff},t} \leq \eta r_{\max}$ by definition. Combining these bounds and using the assumption $r_{\max} \leq \frac{2}{\eta(\mu+L)}$, we can express the range of $\eta_{\text{eff},t}$ as:

$$\min\left\{\frac{2}{\mu+L}, \frac{1}{L(w_t)}\right\} \leq \eta_{\text{eff},t} \leq \frac{2}{\mu+L} \tag{12}$$

This refined bound reveals the adaptive nature of ACSS:

1. In regions of low curvature, where $L(w_t)$ is small, the lower bound $\frac{1}{L(w_t)}$ becomes larger. This allows $\eta_{\text{eff},t}$ to take on larger values, potentially approaching $\frac{2}{\mu+L}$. As a result, ACSS can take larger steps in these flatter regions of the loss landscape.

2. Conversely, in regions of high curvature, where $L(w_t)$ is large, the lower bound $\frac{1}{L(w_t)}$ becomes smaller. This constrains $\eta_{\text{eff},t}$ to smaller values, ensuring that ACSS takes more conservative steps in these highly curved areas of the loss landscape.

Through this mechanism, ACSS naturally adapts its step size to the local geometry of the function, balancing between rapid progress in flat regions and careful navigation in curved regions. $\square$

# C  GENERALIZED ALGORITHM: OPT-ACSS

For any optimizer OPT, we can derive an ACSS version using Algorithm 2. The key modification in the weight and state update steps of the existing optimizer is the substitution of the gradient at time $t$, $g_t$, with $\hat{r}_t g_t / |g_t|$. Using this adaptation, we can incorporate the ACSS mechanism into various optimizers.

---

**Algorithm 2:** Arbitrary optimizer OPT with adaptive curvature step size (OPT-ACSS)

---

**Input:** Function $f : \mathbb{R}^n \times \mathcal{D} \to \mathbb{R}$, initial parameters $w_0 \in \mathbb{R}^n$, base learning rate $\eta$, maximum radius $r_{max}$, number of iterations $T$, batch size $B$, Optimizer parameter update function: UpdateParams, Optimizer weight update function: UpdateWeights

**Output:** Optimized parameters $w_T$

Initialize optimizer state $S_0$ according to the specific optimizer;

**for** $t = 0$ **to** $T - 1$ **do**

    Sample a mini-batch $\mathcal{B}_t$ from $\mathcal{D}$;

    Compute gradient $g_t = \nabla_w f(w_t, \mathcal{B}_t)$ and next point gradient $g'_t = \nabla_w f(w_t - \eta g_t, \mathcal{B}_t)$;

    Compute normalized radius of curvature $r_t = \frac{||g_t||}{||g_t - g'_t||}$;

    Compute capped radius $\hat{r}_t = \min\{r_{max}, r_t\}$;

    Compute ACSS-adjusted gradient $\tilde{g}_t = \hat{r}_t \times \frac{g_t}{||g_t||}$;

    Update optimizer state $S_t = \text{UpdateState}(S_{t-1}, \tilde{g}_t)$;

    Compute update $\Delta w_t = \text{UpdateWeights}(S_t, \tilde{g}_t)$;

    Update parameters $w_{t+1} = w_t + \Delta w_t$;

**end**

**return** $w_T$

---

This generalization allows for integration of ACSS into various existing optimization algorithms such as SGD, Adam, AdaGrad, and RMSProp, enhancing their performance with its curvature-based step size adjustment.

# D  LIMITATIONS

While ACSS offers significant benefits in terms of optimization performance, it's important to acknowledge its primary limitation: increased computational time per iteration. This additional computational cost arises from the need to compute a secondary gradient and perform additional calculations to determine the adaptive step size. To quantify this limitation, we provide both experimental and theoretical evidence of the additional time required by ACSS methods compared to their non-ACSS counterparts.

## D.1  EXPERIMENTAL EVIDENCE

**Wall-Clock Time Experiments:**  To quantify the computational overhead of ACSS methods compared to their non-ACSS counterparts, we conducted comprehensive wall-clock time experiments. Table 3 presents the results of these experiments, focusing on the mean time taken to complete 2 epochs on the IMDB dataset using various optimizers.

These results offer several insights:

1. **Computational Overhead:** As expected, ACSS methods require more computation time than their non-ACSS counterparts. On average, ACSS methods take approximately 1.37 times longer to complete the same number of epochs.

2. **Consistency Across Optimizers:** The overhead ratio is relatively consistent across different optimization algorithms, ranging from about 1.33 to 1.46 times the non-ACSS version's runtime.

3. **Memory Efficiency Trade-off:** While there is a computational time overhead, it's crucial to emphasize that the primary trade-off that the ACSS method provides is in memory efficiency. Our method achieves results equivalent to several second-order methods while maintaining a significantly lower memory footprint.

Table 3: Mean time to complete 2 epochs on the IMDB dataset using various optimizers

| Optimizer | Wall-clock time (Mean) | Wall-clock time (Std Deviation) | Ratio of Times Taken |
|---|---|---|---|
| SimpleSGD | 91.0175 | 5.0617 | |
| SimpleSGDCurvature | 122.5004 | 2.6835 | 1.3459 |
| Adam | 86.4103 | 0.1938 | |
| AdamCurvature | 121.6974 | 1.3342 | 1.4084 |
| HeavyBall | 85.5865 | 0.5449 | |
| HeavyBallCurvature | 120.9661 | 0.1746 | 1.4134 |
| NAG | 85.6665 | 0.1808 | |
| NAGCurvature | 125.0773 | 1.5621 | 1.4600 |
| Adagrad | 88.1545 | 0.5783 | |
| AdagradCurvature | 119.6787 | 0.5171 | 1.3576 |
| Adadelta | 91.4525 | 1.0088 | |
| AdadeltaCurvature | 124.8485 | 0.3255 | 1.3652 |
| RMSProp | 89.4943 | 1.8763 | |
| RMSPropCurvature | 125.4326 | 0.8316 | 1.4016 |
| RMSPropMomentum | 89.9954 | 0.7421 | |
| RMSPropMomentumCurvature | 124.8127 | 0.2511 | 1.3869 |
| AdamW | 89.5067 | 0.8976 | |
| AdamWCurvature | 125.9545 | 4.0044 | 1.4072 |
| NAdam | 91.6765 | 0.1706 | |
| NAdamCurvature | 125.4949 | 1.8784 | 1.3689 |
| NAdamW | 91.1489 | 2.7840 | |
| NAdamWCurvature | 124.9436 | 0.9476 | 1.3708 |
| AMSGrad | 91.5774 | 2.1047 | |
| AMSGradCurvature | 121.5177 | 2.3766 | 1.3269 |

## D.2 THEORETICAL ANALYSIS OF COMPUTATIONAL COMPLEXITY

To complement our empirical results, we provide a theoretical analysis of the computational complexity of ACSS compared to standard SGD.

**Theorem 11** (Computational Complexity of ACSS vs. SGD). Let $f : \mathbb{R}^n \to \mathbb{R}$ be the objective function for a neural network, $n$ be the number of parameters, and $B$ be the mini-batch size. Let $C_{\text{gc}}$ represent the cost of gradient computation per sample per parameter.

The ratio of computational cost per iteration for ACSS vs SGD is approximately 2, assuming $C_{\text{gc}} \gg 1$. In other words:

$$\frac{\text{Cost}_{\text{ACSS}}}{\text{Cost}_{\text{SGD}}} \approx 2 \tag{13}$$

*Proof.* We analyze the computational cost of each step in both SGD and ACSS:

| Operation | Description | Cost (FLOPs) |
|---|---|---|
| $c_1$ | Gradient Computation (SGD & ACSS) | $Bn \cdot C_{\text{gc}}$ |
| $c_2$ | Secondary Gradient Computation (ACSS only) | $Bn \cdot C_{\text{gc}}$ |
| $c_3, c_4$ | Norm Calculation (ACSS only) | $2n + 1$ |
| $c_5$ | Ratio Computation (ACSS only) | $1$ |
| $c_6$ | Gradient Normalization (ACSS only) | $n$ |
| $c_7$ | Parameter Update (SGD & ACSS) | $n$ |

Table 4: Computational cost breakdown for SGD and ACSS operations

Summing up for SGD:
$$\text{Cost}_{\text{SGD}} = c_1 + c_7 = Bn \cdot C_{\text{gc}} + n \text{ FLOPs}$$

Summing up for ACSS:

$$\text{Cost}_{\text{ACSS}} = c_1 + c_2 + c_3 + c_4 + c_5 + c_6 + c_7$$
$$= Bn \cdot C_{\text{gc}} + Bn \cdot C_{\text{gc}} + (2n+1) + (2n+1) + 1 + n + n$$
$$= 2Bn \cdot C_{\text{gc}} + 7n + 3 \text{ FLOPs}$$

The additional overhead of ACSS is therefore:

$$\Delta\text{Cost} = \text{Cost}_{\text{ACSS}} - \text{Cost}_{\text{SGD}}$$
$$= (2Bn \cdot C_{\text{gc}} + 7n + 3) - (Bn \cdot C_{\text{gc}} + n)$$
$$= Bn \cdot C_{\text{gc}} + 6n + 3 \text{ FLOPs}$$

Given that $C_{\text{gc}} \gg 1$ in practice, the dominant term in both algorithms is $Bn \cdot C_{\text{gc}}$. ACSS effectively doubles this term, leading to approximately twice the computational cost of SGD per iteration. $\qquad\square$

This theoretical analysis aligns with our empirical observations, confirming that ACSS introduces a significant but consistent computational overhead compared to standard optimization methods.

In conclusion, while ACSS methods introduce a computational overhead of approximately 1.37 times longer runtime, this is balanced by significant memory efficiency. By providing second-order-like benefits without increasing memory footprint, ACSS offers a valuable alternative for large-scale problems and memory-constrained scenarios. This makes ACSS particularly useful when memory constraints outweigh computational time considerations, introducing a new option for balancing time and memory trade-offs in optimization.

# E  ADDITIONAL EXPERIMENTAL RESULTS

## E.1  COLA DATASET PERFORMANCE:

In our experiments with the CoLA (Corpus of Linguistic Acceptability) dataset, we evaluated the performance of various optimizers with and without the Adaptive Curvature Step Size (ACSS) method over five epochs. The ACSS variants consistently outperformed their traditional counterparts throughout the training process.

RMSProp and RMSProp-ACSS initially performed similarly (0.634 vs 0.636), but by the fifth epoch, the ACSS version significantly outperformed the standard version (0.522 vs 0.611). Adagrad showed more modest improvements with ACSS, yet still consistently outperformed its standard counterpart. Adam-based optimizers (Adam-ACSS, AMSGrad-ACSS, AdamW-ACSS, NAdam-ACSS, NAdamW-ACSS) demonstrated similar performance patterns, starting with slightly higher losses but showing consistent improvement over the epochs. By the fifth epoch, these ACSS variants achieved lower losses (around 0.528-0.534) compared to their non-ACSS counterparts (0.596-0.605).

**Key Takeaways:** Table 5 shows a significant outperformance of the ACSS optimizers where the best performing optimizers have only reached a training loss of 0.591 (Adagrad), whereas eight of the ACSS versions beat this training loss at epoch 5.

Table 5: Training Loss over 10 Epochs for CoLA Dataset with a simplified RNN Model. Notice that many of the best models are ACSS versions. Furthermore, the decrease in training loss is often much higher for the ACSS versions of the optimizer.

| Optimizer Name | Regular Optimizer | | | | | ACSS Version of Optimizer | | | | |
|---|---|---|---|---|---|---|---|---|---|---|
| | Epoch 1 | Epoch 2 | Epoch 3 | Epoch 4 | Epoch 5 | Epoch 1 | Epoch 2 | Epoch 3 | Epoch 4 | Epoch 5 |
| Adadelta | $0.610_{\pm0.00}$ | $0.605_{\pm0.00}$ | $0.601_{\pm0.00}$ | $0.597_{\pm0.00}$ | $0.593_{\pm0.00}$ | $0.684_{\pm0.03}$ | $0.645_{\pm0.01}$ | $0.627_{\pm0.00}$ | $0.619_{\pm0.00}$ | $0.616_{\pm0.00}$ |
| Adagrad | $0.611_{\pm0.00}$ | $0.608_{\pm0.00}$ | $0.604_{\pm0.00}$ | $0.599_{\pm0.01}$ | $0.591_{\pm0.01}$ | $0.613_{\pm0.00}$ | $0.601_{\pm0.00}$ | $0.596_{\pm0.00}$ | $0.591_{\pm0.00}$ | $0.588_{\pm0.00}$ |
| Adam | $0.611_{\pm0.00}$ | $0.608_{\pm0.00}$ | $0.605_{\pm0.00}$ | $0.603_{\pm0.00}$ | $0.596_{\pm0.01}$ | $0.620_{\pm0.00}$ | $0.600_{\pm0.00}$ | $0.583_{\pm0.00}$ | $0.560_{\pm0.00}$ | $0.528_{\pm0.01}$ |
| AdamW | $0.611_{\pm0.00}$ | $0.608_{\pm0.00}$ | $0.606_{\pm0.00}$ | $0.602_{\pm0.01}$ | $0.597_{\pm0.01}$ | $0.620_{\pm0.00}$ | $0.600_{\pm0.00}$ | $0.583_{\pm0.00}$ | $0.560_{\pm0.00}$ | $0.528_{\pm0.01}$ |
| AMSGrad | $0.611_{\pm0.00}$ | $0.608_{\pm0.00}$ | $0.606_{\pm0.00}$ | $0.602_{\pm0.00}$ | $0.596_{\pm0.01}$ | $0.620_{\pm0.00}$ | $0.600_{\pm0.00}$ | $0.583_{\pm0.00}$ | $0.560_{\pm0.00}$ | $0.528_{\pm0.01}$ |
| HeavyBall | $0.624_{\pm0.00}$ | $0.611_{\pm0.00}$ | $0.610_{\pm0.00}$ | $0.610_{\pm0.00}$ | $0.609_{\pm0.00}$ | $0.621_{\pm0.00}$ | $0.608_{\pm0.00}$ | $0.603_{\pm0.00}$ | $0.599_{\pm0.00}$ | $0.595_{\pm0.00}$ |
| NAdam | $0.612_{\pm0.00}$ | $0.609_{\pm0.00}$ | $0.608_{\pm0.00}$ | $0.605_{\pm0.00}$ | $0.602_{\pm0.01}$ | $0.623_{\pm0.00}$ | $0.606_{\pm0.00}$ | $0.592_{\pm0.00}$ | $0.569_{\pm0.01}$ | $0.534_{\pm0.01}$ |
| NAdamW | $0.611_{\pm0.00}$ | $0.609_{\pm0.00}$ | $0.608_{\pm0.00}$ | $0.605_{\pm0.00}$ | $0.605_{\pm0.00}$ | $0.623_{\pm0.00}$ | $0.606_{\pm0.00}$ | $0.592_{\pm0.00}$ | $0.568_{\pm0.01}$ | $0.534_{\pm0.01}$ |
| NAG | $0.624_{\pm0.00}$ | $0.611_{\pm0.00}$ | $0.610_{\pm0.00}$ | $0.610_{\pm0.00}$ | $0.609_{\pm0.00}$ | $0.621_{\pm0.00}$ | $0.608_{\pm0.00}$ | $0.603_{\pm0.00}$ | $0.599_{\pm0.00}$ | $0.595_{\pm0.00}$ |
| RMSProp | $0.634_{\pm0.02}$ | $0.617_{\pm0.01}$ | $0.614_{\pm0.01}$ | $0.611_{\pm0.00}$ | $0.611_{\pm0.00}$ | $0.636_{\pm0.00}$ | $0.602_{\pm0.00}$ | $0.584_{\pm0.00}$ | $0.557_{\pm0.01}$ | $0.522_{\pm0.01}$ |
| RMSPropMomentum | $0.635_{\pm0.02}$ | $0.626_{\pm0.04}$ | $0.614_{\pm0.01}$ | $0.612_{\pm0.00}$ | $0.610_{\pm0.00}$ | $0.638_{\pm0.00}$ | $0.604_{\pm0.00}$ | $0.587_{\pm0.01}$ | $0.561_{\pm0.01}$ | $0.525_{\pm0.02}$ |
| SimpleSGD | $0.662_{\pm0.01}$ | $0.630_{\pm0.00}$ | $0.622_{\pm0.00}$ | $0.618_{\pm0.00}$ | $0.616_{\pm0.00}$ | $0.611_{\pm0.00}$ | $0.608_{\pm0.00}$ | $0.606_{\pm0.00}$ | $0.605_{\pm0.00}$ | $0.603_{\pm0.00}$ |

## F  Details of Testing Functions for ACSS Optimization

We provide details on the four functions used to test the ACSS based optimizer below.

### F.1  The Rosenbrock Function

The function is depicted with contour lines, where darker colors indicate lower values. Each subplot displays the path taken by a different optimizer. The plots indicate that the ACSS versions of the optimizers navigate the function's characteristic narrow, parabolic valley more effectively, by reducing the step size as appropriate. The learning rate is set to $1.5 \times 10^{-3}$, and the iterates start at $(-1.5, 2)$.

### F.2  The Easom Function

The Easom function features a broad, flat area with a sharp depression at its global minimum $(\pi, \pi)$. With a learning rate of $2.0 \times 10^{-3}$ and 200 iteration steps, standard optimizers remain near the initial point. In contrast, ACSS versions achieve convergence, showing ACSS's capability to accelerate optimization in low-gradient scenarios.

### F.3  The Ackley Function

The Ackley function presents a flat outer region with numerous local minima and a steep central hole containing the global minimum at (0,0). With a learning rate of $5 \times 10^{-3}$ and 25 iterations, ACSS versions of optimizers demonstrate superior navigation of the loss landscape, adaptively reducing step size near convergence.

### F.4  The Three-humped Camel Function

The Three-Hump Camel function has three local minima and a global minimum at (0, 0). Using $1.0 \times 10^{-2}$ learning rate for 300 steps, Heavyball and Nesterov methods overshoot, while ACSS versions self-correct, showing enhanced optimization in this complex landscape.

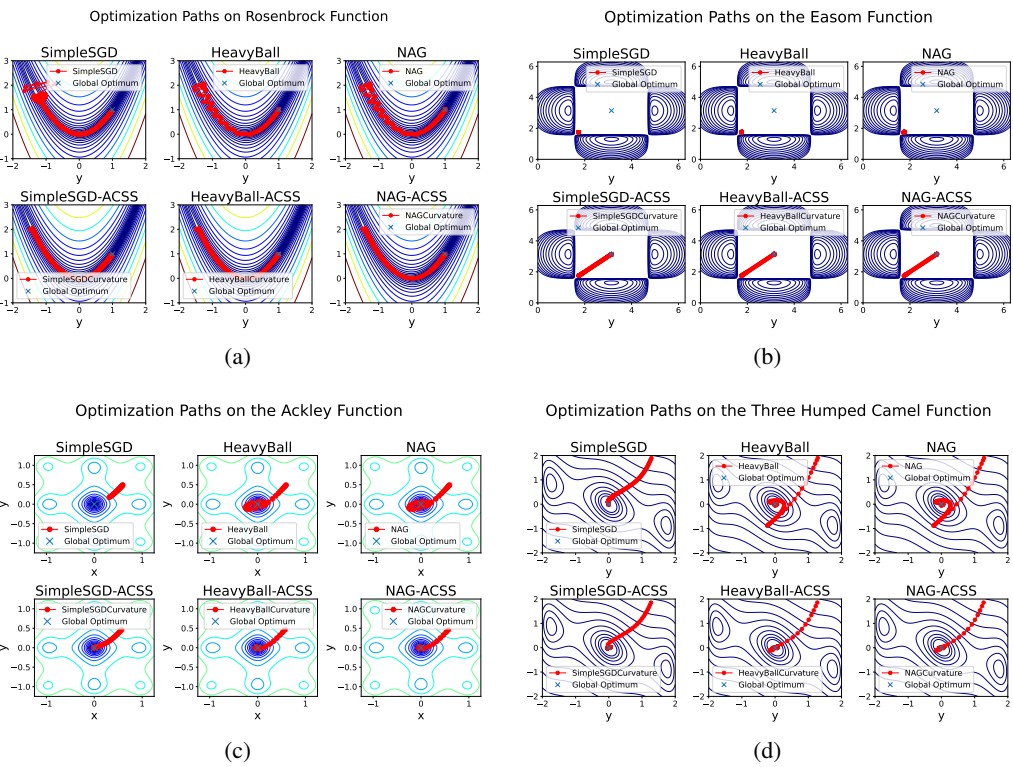

Figure 8: Optimizer performance on challenging optimizer benchmarking functions.