# OpenReview forum: "Adaptive Curvature Step Size: A Path Geometry Based Approach to Optimization"
_ICLR.cc/2025/Conference — ICLR 2025 Conference Withdrawn Submission_

### Official Review · Reviewer_bHyW · 2024-10-23

**Soundness:** 2
**Presentation:** 3
**Contribution:** 2
**Rating:** 5
**Confidence:** 3

**Summary:**

The author(s) present a way to choose step sizes that incorporate the rate of change in the gradient if a plain gradient step were chosen. The intuition is that such choice __adapts__ to the local curvature of the function. The theoretical analysis they perform focuses on $\mu$ strongly convex and $L$ smooth functions, for which they derive the interval of step-sizes that ensure convergence. Under the same assumptions, they are able to derive a convergence rate that is "almost" classical, and also that gradients exhibit some stability wrt perturbations, suggesting that a stochastic setting would not hurt the results. Finally, they quickly demonstrate how the update rule is almost invariant to rescalings. On the experimental side, a large collection of comparisons of classical algorithms and algorithms enhanced with their method is proposed.

Overall I feel like it is a sufficient conference paper: a cute idea that takes up just the right space in terms of pages.

**Strengths:**

The main strength is the presentation and clarity of the idea and the effort in validation. Details in the appendix are possibly even too extensive, the cost of computing all these experiments, whether in time or in computational resources, must have been substantial. The idea is simple enough, and checked over synethtic settings, notoriously hard to optimize functions and realistic datasets. The theoretical validation is very classic: a good starting point to maybe think of extending to more generic functions.
Also good that you recognize in the paper that one limitation is the increased cost of computation.

**Weaknesses:**

- improvement in realistic datasets is very slight in many cases when considering "industrial" optimizers;
- figure 5 shows no evidently better performance of the method, or at least it is not immediate from sight: this is also because validation is over a grid that is too fine (too many cells), so seeing is difficult.
- figure 7: both methods seem to get to the minimum, just that without ACSS the path is more "chaotic".
- I know that the strong convexity-smoothness assumption is standard, but is it that the method converges only under these assumptions (hence potentially limited), or that there is room for extension? Same questions for stochasticity.

###### Typos
Please do not count these as weaknesses.
- the submission is strange. The main text only has main text, and the supplementary material has the main text + the appendix. I guess this is non-standard(?) At least I never saw something like this and maybe it is a mistake.
- abstract line 025 repeats twice the same sentence.
- denominator in line 866-867: Did you miss a $-\epsilon$?

**Questions:**

- is table $1$ really needed in the main text?
- line (137-138), the term "often" is very much not quantitative in this context.
- How many epochs are done in Figures 2, 3
- the mean in Fig. 4 is for how many samples? How many epochs?
- how many epochs in figure 6
- do you have any comments on the approximation you make in line 593? I agree that this does not impact your theoretical results, but how does it influence them? Possibly, this is a rightful comment to make instead of using up space with Table 1.
- the comment about the local-Lipschitz constant is nice, but the constant depends not only on $w_t$, but also on the stepsize $\eta$ I think, right?

---

> ### Comment · Reviewer_bHyW · 2024-11-17
> **Adjustment of scores after reading reviews**
>
> Dear author(s), after reading the other reviewers' comments, I am adjusting my score.
> I would have given it a 6. The score of 8 was in the end kind of imprecise due to ICLR not providing many choices.
>
> However, I have to stress that I will vote for rejection if the proof and code issues of the other reviewer are not addressed. Thanks to MiiW for doing a very heavy review.

---

> > ### Comment · Reviewer_bHyW · 2024-11-25
> > **Score Update**
> >
> > Dear author(s),
> > given the absence of defense against the comments of the other reviewer, which I checked, I lean for rejection.
> >
> > My score update is "marginally below", just because 3 on a scale of 10 is too harsh given my initial review and my personal comments.

---

### Official Review · Reviewer_MiiW · 2024-10-28

**Soundness:** 1
**Presentation:** 2
**Contribution:** 2
**Rating:** 3
**Confidence:** 5

**Summary:**

This work proposes a new general framework called Adaptive Curvature Step Size (ACSS) for optimization algorithms. It first captures the local geometry of the loss landscape by computing the “normalized radius of curvature” (which takes two forward/backward passes per iteration) and uses it to adjust the step size automatically. The paper claims that it provides theoretical guarantees on ACSS. Lastly, ACSS is empirically tested on several deep-learning tasks with various datasets and architectures.

**Strengths:**

S1. This work presents a novel idea of automatically rescaling the step size based on the local geometry.

S2. Among theoretical analyses, Theorem 4 states the scale (near-)invariance in terms of the step size. I agree that this brings a benefit in practice because it reduces the burden of hyperparameter tuning.

S3. The authors conducted extensive empirical evaluations of their method to prove that ACSS is beneficial for fast convergence on the training loss.

**Weaknesses:**

I found the soundness of the paper is not good enough, mainly because of (1) the weak or wrong theory and (2) the wrong PyTorch implementation. I list the weaknesses below, including these two and more.

**W1. Gradient Descent with ACSS (GD-ACSS) practically does NOT converge to the minimum, even on a strongly convex $L$-smooth function.**

- Theorem 1 implies that the effective step size is bounded above and below by positive values. By following the author’s proof of Theorem 5 (which does not require strong convexity in essence), it is easy to deduce that the effective step size $\eta_{\rm eff}$ is bounded as \\[ \min \left\\{ \frac{1}{L}, \eta r_{\rm max} \right\\} \le \eta_{\rm eff} \le \eta r_{\rm max}. \\] Now let me denote $Q := \min \left\\{ \frac{1}{L}, \eta r_{\rm max} \right\\}$, the lower bound of the effective step size.
- The update rule of GD-ACSS (Equation 4) implies that the effective step size is identical to the distance between two consecutive iterates: $\eta_{\rm eff} = \\| w_{t+1} - w_t \\|$. Thus, by Theorem 1, every two consecutive iterates are far away at least $Q>0$.
- This contradicts Theorem 2, the linear convergence result of GD-ACSS on a strongly convex $L$-smooth function. If Theorem 2 were true, by triangle inequality, the non-vanishing term can be bounded above as \\[ \\| w_{t+1} - w_t \\| \le \\| w_{t} - w^* \\| + \\| w_{t+1} - w^* \\| \le 2 \cdot \left( 1 - \frac{\mu^2}{L^2} \right)^{t/2} \\| w_0 - w^* \\|. \\] However, the R.H.S. of the inequality above must vanish to zero as $t\rightarrow\infty$.
- Which one is correct: Theorem 1 or Theorem 2? Indeed, Theorem 2 is wrong. One can easily prove the non-convergence by letting $f(w) = \frac{L}{2} w^2$: the update rule becomes $w_{t+1} = w_t - Q \cdot {\rm sign}(w_t)$. Let $w_0=mQ + r$, where $m$ is an integer and $r\in (0, Q)$ (i.e., $w_0$ is not an integral multiple of $Q$). Then, since $|w_{t+1} - w_t| = Q$, there exists an integer $k_t$ such that $w_t = k_t Q + r$ for every $t$. Thus, the minimum possible distance from $w^*=0$ to $w_t$ is $\min\\{r, Q-r\\} > 0$ .
- Then, where is the proof of Theorem 2 (or Theorem 9 in Appendix) wrong? The “proof” tries to obtain a “lower bound” of the contraction factor in terms of $\\| w_{t} - w^* \\| $. However, to prove the convergence upper bound, you must obtain the “upper bound” of the contraction factor over every time step $t$. The authors probably wanted to make the contraction factor smaller to optimize the hyperparameters (e.g., step size), but that was not the case in the given proof.
- I also checked the non-convergence through PyTorch code. Although the author’s original implementation was a bit erroneous (see **W5** for the details), I fixed it on my own. Then, I tested GD-ACSS on a quadratic loss $f(x,y) = \frac{1}{2} x^2  + \frac{0.01}{2} y^2$. I used `lr=0.001`, `epsilon=0`, `clip_radius=2/(lr*(1+0.01))` (following the choice of $r_{\rm max}$ in Theorem 1), and the initialization at $(10,10)$. As a result of running GD-ACSS for 20000 iterations, I found its later iterates oscillate between two points near $(0, -0.8)$ and $(0, 0.2)$, and the loss does not decrease below $0.01$.

**W2. Other comments on theoretical analyses**

- Strictly speaking, Theorem 1 is also erroneous because, as I mentioned above, the lower bound of the effective step size must be $ \min\left\\{ \frac{1}{L}, \eta r_{\rm max} \right\\}$ instead of just $\frac{1}{L}$, because $r_{\rm max}$ can be smaller than $\frac{1}{\eta L}$. In fact, the fifth equation in the proof of Theorem 5 is wrong: it is true only when $r_{\rm max} \ge \frac{2}{\mu+L}$, which is different from the assumption.
- Theorems 6 and 7 are about Gradient Descent without ACSS. It is confusing to use the symbol $\eta_{{\rm eff}, t}$ to denote the (constant) step size of Gradient Descent. Also, I don’t understand the purpose of these theorems. Is this for comparison with GD-ACSS? Does the paper compare them in terms of convergence rate?
- Lemma 1 itself is not true in general. $f$ must be twice-differentiable (which cannot be implied by L-smoothness nor strong convexity); otherwise, the statement of Lemma 1 is true for “almost any” $\xi$ (by Rademacher’s Theorem).
- Theorem 3 (or, equivalently, Theorem 8) is about the stability of GD-ACSS. I cannot understand why the authors put this result. What does the stability in terms of gradient perturbation imply? Moreover, I am quite suspicious of the correctness of its proof in many ways: I cannot understand how the triangle inequalities are applied and what happens if $\epsilon < m < 2\epsilon$ (there is no assumption about the relationship between $m$ and $\epsilon$).
- In most of the proofs, the gradient boundedness (i.e., Lipschitz continuity) and strong convexity are assumed together, although they are not actually applied at once in each proof. In general, however, these two assumptions cannot hold if the domain is unbounded, because a Lipschitz continuous and strongly convex function must have a convex bounded domain. The gradient boundedness along the optimization trajectory must be proved (not assumed) if the authors want to use it.
- Lastly, there are no pointers to the proofs in the main text, which makes it difficult to read the paper.

**W3. Regarding memory consumption**

- Unlike the author’s claim, ACSS seems to require additional memory consumption, not only taking additional computational burden (admitted by the authors in Section 4.7). This is mainly because two different gradient vectors, $g_t = \nabla f(w_t)$ and $g’_t =  \nabla f(w_t- \eta g_t)$, must be stored. Moreover, we need to store both $w_t$ and $w’_t = w_t - \eta g_t$ to perform forward-/backward-passes on these two weight vectors. Thus, I guess the ACSS requires twice as much memory and computation compared to the base optimizer. Please correct me if I am wrong. It would be great to numerically compare the memory consumption with or without ACSS.

**W4. Regarding comparisons of the optimization algorithms**

- I am concerned about the fairness of comparisons among the optimization algorithms. As I mentioned in **W3** and recognized by the authors, ACSS requires twice as many gradient computations per iteration. However, for every empirical result in this paper, the authors compare the regular optimizers and their ACSS-based counterparts by running them for a fixed number of epochs. I don’t think this is a fair comparison. Also, if we compare the training losses at Epoch $2n$ (for a regular optimizer) and at Epoch $n$ (for an ACSS-based counterpart), the improvement becomes less impressive. For example, the authors claimed that AdamW-ACSS significantly improves the performance over AdamW. However, if we compare Epoch 4 (for AdamW) and Epoch 2 (for AdamW-ACSS), AdamW seems much better than its ACSS counterpart (0.499 < 0.589).
- In addition to the fairness of the comparisons, it is also questionable that the training loss is used for measuring the performance of the optimizers throughout the paper. In the introduction section, the authors motivate the readers about the importance of new optimization techniques in the context of deep learning (DL). However, due to the large expressiveness of deep and large models, reducing the training loss to near zero is often regarded as an easy problem. Arguably, what really matters in the DL context is to reduce the test (or validation) error and to achieve successful generalization. In this context, I wonder why the authors chose to exhibit only the training performances, rather than the generalization performances, in their paper.

**W5. The author’s PyTorch implementation seems WRONG.**

- I guess I found a couple of errors or misleading codes in the author’s PyTorch implementations.
- First of all, the implemented ACSS-based optimizers do not compute forward-/backward-passes at $w_t - \eta g_t$. Instead, it does the same computation on $w_t$ twice. The method `step()` of an optimizer instance computes the loss twice and there are two loops for gradient updates. In the first loop, however, it only “computes” the gradient but does not update the current parameter $w_t$ to $w_t - \eta g_t$. Hence, the second forward pass computes exactly the same loss as the one before the first loop (and thus `current_grad` is the same as `last_grad`). This is why we immediately get NaN if we set `epsilon=0` in the original implementation (because `current_grad-last_grad` is always zero).
- On top of that, except for SimpleSGD, NAG, and HeavyBall, every other ACSS-based optimizer implementation does not contain the parameter `clip_radius`, corresponding to $r_{\rm max}$. Instead, every ACSS-based implementation has `epsilon`. It corresponds to $\epsilon>0$ of a modified formula for “normalized radius of curvature”: \\[r_t = \frac{\\|g_t\\|}{\max\\{\\|g_t - g’\_t\\|, \epsilon\\}}.\\] This is clearly different from the explanation in Section 2 of the paper. This is because, with the formula above and without clipping radius $r_{\rm max}$, the effective step size $\eta_{\rm eff}:=\eta {r}_t$ is no longer lower bounded by a positive value as illustrated in Theorem 1; instead, it can be zero when $g_t = 0$. This is a vast difference because it may allow the convergence, unlike what I argued in **W1**. I guess the successful convergences showcased in Figures 1, 7, and 8 are due to the presence of the parameter `epsilon`, but it is not explicitly explained in the main text.

**W5. Minor typos and writing issues**

- The abstract seems to require a re-writing. It unnecessarily repeats (three times!) the phrases/clauses of basically the same idea about adaptation to local loss landscape: “[…], which dynamically adjusts the step size based on the local geometry of the optimization path”, “The effectiveness of ACSS stems from its ability to adapt to the local landscape of the optimization problem”, and “A key advantage of ACSS is its adaptive behavior based on local curvature information”.
- Line 88 or 89: duplicate word ‘through’. Also, the citation “Loshchilov & Hutter (2017)” must be inside `\citep` environment instead of `\citet`.
- Section 2.1: It should contain a pointer to Appendix C, which includes the general algorithmic framework of applying ACSS to the existing optimization algorithm (OPT-ACSS).
- Section 4.2: It contains no pointer to Table 2.
- Lines 748 and 754: ‘L’ of ‘L-smooth…’ should be inside a math mode: ‘$L$-smooth’.
- Proposition B.1: the numbering is inconsistent with other statements like Theorems and Lemmas.

**Questions:**

**Q1. (Im)possibility of escaping the sharp & bad local minima**

- Although I recognize the soundness of the idea of autonomously adapting the step size based on the local geometry, I am worried about the bad local minima problem. If ACSS takes small steps in regions of high curvature, doesn’t ACSS struggle to escape from a sharp (i.e., high curvature) local minima? If it does, ACSS might not be desirable for training modern deep models (especially when its effective condition number is extremely high).

**Q2. Insensitivity to the directional change of the local gradients**

- I found the definition of the normalized radius of curvature $r_t$ ignores the local change in gradient directions. For instance, $r_t$ remains the same if $g’_t = 3g_t$ or $g’_t = -g_t$. I think this directional change is also crucial for estimating the local geometry. Can we do better if we incorporate this local directional change of gradients? Do you have any guesses?

---

### Official Review · Reviewer_tKN1 · 2024-10-29

**Soundness:** 2
**Presentation:** 2
**Contribution:** 3
**Rating:** 5
**Confidence:** 4

**Summary:**

The paper proposes the Adaptive Curvature Step Size (ACSS), a novel optimization method that adjusts step sizes dynamically based on the local curvature of the optimization path. The method estimates the radius of curvature using consecutive gradients and sets the step size proportionally. ACSS aims to capture second-order information without requiring additional memory storage, making it a computationally efficient alternative to traditional optimization techniques. The authors evaluate ACSS across various datasets and optimizers, demonstrating improvements in both training efficiency and performance.

**Strengths:**

* The proposed method introduces a novel approach for capturing second-order information without requiring additional storage.
* The authors demonstrate that the new method achieves linear convergence under strongly convex and smooth conditions.
* Comprehensive experiments conducted by the authors show that the proposed method improves upon existing algorithms in many cases, particularly those that do not exploit second-order information.

**Weaknesses:**

* The theoretical analysis is limited to the strongly convex, smooth, and deterministic setting, and it demonstrates the same convergence rate as standard gradient methods. This makes it difficult to convincingly argue the superiority of the proposed ACSS method based on the theoretical results alone.
* The experimental results are based on training for a fixed number of epochs, which may not reflect the ultimate performance of each algorithm. This leaves me uncertain about the claimed superiority of the proposed method.
* The experimental setup is not described in sufficient detail, and it is unclear whether the hyperparameters for each algorithm were appropriately selected, which raises concerns about the fairness of the comparison.
* The figure design is somewhat misleading, as blue is used to indicate improvement in Figure 2, while green is used for the same purpose in Figure 3. Additionally, providing the average rankings in Figure 5 would offer a more straightforward comparison.

**Questions:**

* From equations (2) and (3) in the paper, it appears that the effect of the step size is canceled out, as also shown in Theorem 4. Could the authors provide further justification for this choice?
* In Theorem 3, why is a lower bound on the gradient norm necessary?

---

### Official Review · Reviewer_26q5 · 2024-10-31

**Soundness:** 3
**Presentation:** 3
**Contribution:** 2
**Rating:** 5
**Confidence:** 3

**Summary:**

This paper proposes the Adaptive Curvature Step Size (ACSS) method, an optimization approach that dynamically adjusts step size based on the curvature of the optimization path. By estimating the local curvature using consecutive gradients, ACSS provides a normalized radius to set the step size, enabling larger steps in low-curvature regions and smaller, more cautious steps in high-curvature regions. The method integrates with common optimizers like SGD, Adam, and RMSProp, providing second-order benefits without additional memory overhead. Extensive experiments demonstrate that ACSS enhances optimization efficiency across 20 diverse datasets, with performance gains over various popular optimizers.

**Strengths:**

1. The approach introduces a novel use of path geometry for step size adaptation, which leverages the optimization path’s curvature to inform step adjustments dynamically.

2. The paper clearly outlines the algorithm, the derivation of key formulas, and includes theoretical and empirical analyses to support the effectiveness of ACSS.

**Weaknesses:**

1. ACSS introduces additional computational costs since it requires twice backpropagation per iteration.

2. Given that geometry-inspired methods like Sharpness-Aware Minimization (SAM) have shown benefits for generalization, it would be valuable for the authors to explore whether ACSS similarly improves test performance. Including these results could enhance the understanding of ACSS’s applicability to broader model generalization.

3. The paper lacks a detailed sensitivity analysis of the ACSS hyperparameters. Understanding the robustness of the approach to these values would be valuable for general applicability.

**Questions:**

1. Could the authors provide additional guidance on how practitioners should interpret and set values for r_t and r_{\text{max}} in practical scenarios? How sensitive is the performance of ACSS to these values?

2. The paper primarily focuses on optimization error in training but does not report test performance. Recent research, such as Sharpness-Aware Minimization [1], which is also geometry-inspired and involves additional backpropagation steps, has demonstrated benefits for generalization. Could the authors discuss whether ACSS might similarly benefit generalization, particularly in terms of test performance? Including results or a discussion on generalization would provide a more comprehensive view of ACSS’s effectiveness beyond training optimization.

3. Have the authors considered integrating a technique similar to LookSAM [2], which uses historical gradient information to approximate sharpness adjustments without an additional backward pass, to potentially reduce the computational cost of ACSS? Leveraging historical gradients might preserve the adaptive benefits of ACSS while improving its efficiency?

[1] Sharpness-Aware Minimization for Efficiently Improving Generalization

[2] Towards Efficient and Scalable Sharpness-Aware Minimization

---

### Official Review · Reviewer_EAgB · 2024-11-02

**Soundness:** 4
**Presentation:** 4
**Contribution:** 3
**Rating:** 6
**Confidence:** 2

**Summary:**

This paper introduces an approach to dynamically adjust the step size during optimization based on the curvature of the loss landscape called ACSS. ACSS calculates the normalized radius of curvature using current gradient $g_t$ and “tentative next point gradient” $g_t*$ and then calculates the curvature $r$ using $\frac{|| g_t || }{ || g_t - g_t* || }$. The calculated curvature $r$ is then used as step size. This allows it to take larger steps in flat regions and smaller steps in regions with high curvature, navigating complex loss landscapes efficiently. The authors then provide a theoretical analysis showing that the step size is bounded, and their method achieved linear convergence for strongly convex function. They also show that for strongly convex, smooth functions, ACSS is stable under gradient permutations and is scale invariant. Empirically evaluation on 20 datasets and across 12 different optimizers showcases the applicability of ACSS across different datasets and optimizers. The excellent summarization and visualization of experiments clearly showcases that for a given epoch, methods with ACSS tend to perform better than those without.

**Strengths:**

-- The paper introduces a novel approach to dynamically adjust the step size during optimization based on the curvature of the loss landscape called ACSS. I like the simplicity of the approach.

-- The presentation of the paper is excellent. The paper is very well written, easy to follow and a very good read.

-- Empirical results presented in the paper are very thorough. They have evaluated their method across 12 different optimizers and 20 different datasets. Their thorough comparisons clearly showcases the effectiveness of ACSS.

-- Theoretically again for smooth and strongly convex functions they provide a clear and thorough analysis showcasing the usefulness of their adaptive step size proposal.

**Weaknesses:**

1) Practical Consideration leading to incomplete experiments (IMO): For large models with Billions of parameters, the gradient function is very resource intensive to compute (both in memory and in compute). The Algorithm 1 computes the current gradient (which is true for all SGD algorithm) and *tentative gradient at the next point*.

a) If my understanding is correct this will double the memory and compute required for each epoch? This can lead to OOM issues for large parameter models.

b) The experiment section (though thorough in the list of algorithms and datasets it compares to) only compares epoch wise and shows gains and losses accordingly. What about the peak memory requirement and the total compute /flops required either per epoch or per dataset to get to the optima?

c) Instead of using equation 2, $\frac{|| g_t || }{ || g_t - g_t* || }$ have you thought about approximations either in the form of $g_t* = g_{t-1}$ or some sort of taylor series (which might lead to hessian which can be approximated by first order methods)

I will request the authors to respond to this question in as much detail as possible.

2) Significance of the theoretical analysis. I am unclear about the significance of the theoretical analysis section. For example, given that the step size is bounded, for a smooth and strongly convex function the effective stepsize will be bounded (theorem 1). It will be helpful to hear more on the theoretical significance of theorems.

**Questions:**

Overall, I have a slight positive inclination towards the paper but would want the authors to still answer the questions raised in the weakness section.

---

### Note · Authors · 2024-12-01

**Comment:**

We need more time to amend our submission in line with the reviewer comments. While the optimization method works well, we have not done justice to it in this submission. Withdrawing the paper at this time to do better justice to the method, specifically to correct some errors in the Theorem statements.

Thanks to all the reviewers for their time reviewing our work.

**Withdrawal Confirmation:**

I have read and agree with the venue's withdrawal policy on behalf of myself and my co-authors.